# Differential regulation of the *Drosophila* sleep homeostat by circadian and arousal inputs

Jinfei D Ni[1,2,3], Adishthi S Gurav[1,3], Weiwei Liu[1,3], Tyler H Ogunmowo[1,3], Hannah Hackbart[1,3], Ahmed Elsheikh[1,3], Andrew A Verdegaal[1,3], Craig Montell[1,3]*

[1]Department of Molecular, Cellular and Developmental Biology, University of California, Santa Barbara, Santa Barbara, United States; [2]Department of Biological Chemistry, The Johns Hopkins University School of Medicine, Baltimore, United States; [3]Neuroscience Research Institute, University of California, Santa Barbara, Santa Barbara, United States

**Abstract** One output arm of the sleep homeostat in *Drosophila* appears to be a group of neurons with projections to the dorsal fan-shaped body (dFB neurons) of the central complex in the brain. However, neurons that regulate the sleep homeostat remain poorly understood. Using neurogenetic approaches combined with $Ca^{2+}$ imaging, we characterized synaptic connections between dFB neurons and distinct sets of upstream sleep-regulatory neurons. One group of the sleep-promoting upstream neurons is a set of circadian pacemaker neurons that activates dFB neurons via direct glutaminergic excitatory synaptic connections. Opposing this population, a group of arousal-promoting neurons downregulates dFB axonal output with dopamine. Co-activating these two inputs leads to frequent shifts between sleep and wake states. We also show that dFB neurons release the neurotransmitter GABA and inhibit octopaminergic arousal neurons. We propose that dFB neurons integrate synaptic inputs from distinct sets of upstream sleep-promoting circadian clock neurons, and arousal neurons.

DOI: https://doi.org/10.7554/eLife.40487.001

*For correspondence:
cmontell@ucsb.edu

Competing interests: The authors declare that no competing interests exist.

## Introduction

Many of the regulatory mechanisms controlling sleep have been deciphered in model organisms such as flies and mice, and appear to be conserved in humans (*Borbély, 1982*; *Griffith, 2013*; *Harbison et al., 2009*; *Sehgal and Mignot, 2011*; *Shaw et al., 2000*; *Tomita et al., 2017*; *Weber and Dan, 2016*). These include dual regulation of sleep by the circadian clock and by homeostatic-drive, both of which are essential for maintaining regular sleep patterns (*Allada et al., 2017*; *Borbély, 1982*). In *Drosophila*, a group of neurons which project to the dorsal layer of the fan-shaped body (dFB neurons) of the central complex in the brain is proposed to be one effector component of the sleep homeostat (*Donlea et al., 2014*; *Donlea et al., 2011*; *Liu et al., 2016*). These dFB neurons receive inputs from Ellipsoid body R2 neurons, which encode sleep drive (*Liu et al., 2016*) and suppress movements stimulated by sensory cues (*Donlea et al., 2018*). Other non-central complex neurons are also involved in sleep regulation and different neuronal pathways elicit distinct effects on the sleep homeostat (*Seidner et al., 2015*). Nevertheless, circadian pacemaker neurons that directly synapse onto dFB neurons have not yet been reported.

Sleep is essential for animal survival but comes with a large trade-off—other critical behaviors such as feeding, mating and defense only take place when an animal is awake (*Griffith, 2013*). Therefore, arousal-promoting molecules and neurons that stimulate wakefulness provide an essential balance. In *Drosophila*, various arousal signals have been identified and their functions in promoting

wakefulness are conserved in mammals (*Crocker and Sehgal, 2008*; *Liu et al., 2012*; *Sehgal and Mignot, 2011*; *Ueno et al., 2012b*). These include biogenic amines (dopamine and octopamine) and neuropeptides such as Pigment Dispersing Factor (PDF) (*Andretic et al., 2008*; *Crocker and Sehgal, 2008*; *Crocker et al., 2010*; *Parisky et al., 2008*). Previous studies have demonstrated that dopamine can downregulate the activity of the dFB neurons (*Pimentel et al., 2016*). Nevertheless, the anatomy of direct synaptic connections between dopaminergic neurons and dFB neurons is uncharacterized. It remains unknown whether different upstream neurons that regulate the activity of dFB neurons in opposite directions, synapse onto the same or distinct neurons in the dFB. More-over, the behavioral consequences of simultaneous hyperactivation of sleep-promoting and wake-promoting neuronal circuits have not been investigated in animal models. An investigation of sleep behavior under such conditions may provide insights into sleep disorders due to uncoordinated activities of different sleep regulatory neurons.

Here we explore the sleep-regulatory neurons that function upstream of dFB neurons. We identified two groups of sleep-promoting neurons that form glutaminergic excitatory synaptic connections onto dFB neurons. One group of sleep-promoting neurons are circadian pacemaker neurons that also express the neuropeptide, Allatostatin-A (AstA), a sleep-promoting factor as well as a feeding regulator (*Chen et al., 2016*; *Donlea et al., 2018*; *Hergarden et al., 2012*). Additionally, we characterized synaptic connections from upstream dopaminergic arousal neurons onto the dFB neurons and revealed a novel wake-promoting mechanism by which dopamine released from arousal neurons inhibits synaptic output at the axonal terminals of dFB neurons. Lastly, we found that dFB neurons promote sleep by releasing GABA and inhibiting previously identified octopaminergic arousal (OAA) neurons (*Crocker et al., 2010*), thereby conferring long sleep bouts at night. Our results indicate that dFB neurons integrate different signals reflecting the circadian clock, arousal, as well as sleep drive. These signals originate from distinct populations of upstream neurons and converge on and influence the activity of common dFB neurons.

## Results

### Circadian (LPN) and SLP neurons promote sleep and function upstream of the sleep-homeostat

To identify sleep-regulatory neurons that might function upstream of dFB neurons we employed the *Drosophila* Activity Monitoring system and scored sleep as inactivity for a minimum of 5 min, as previously described (*Pfeiffenberger et al., 2010a*; *Pfeiffenberger et al., 2010b*; *Shaw et al., 2000*). To assess the effects of chronically or acutely hyperactivating different classes of neurons, we used the *Gal4/UAS* system to express the depolarization-activated $Na^+$ channel (NaChBac) (*Nitabach et al., 2006*), or the red-shifted channelrhodopsin (CsChrimson) (*Klapoetke et al., 2014*). Control flies harboring *UAS-NaChBac* without any *Gal4* display typical sleep patterns characterized by minimal sleep at dawn and dusk (near ZT0 and ZT12, respectively), which rises to high levels during the middle of the day (near ZT6; *Figure 1A and E*). At night (ZT12—24), the control flies exhibit nearly maximal levels of sleep, which is 720 min (*Figure 1A and F*).

Hyperactivation of dFB neurons using *UAS-NaChBac* and the *23E10-Gal4* (*Donlea et al., 2018*; *Pimentel et al., 2016*), which includes an enhancer from the *Allatostatin-A receptor 1* (*AstA-R1*) gene, greatly increases daytime sleep to levels that normally occur at night (*Figure 1B,D,E and F*). Consistent with a previous report (*Chen et al., 2016*), we found that hyperactivation of neurons with *UAS-NaChBac* and *65D05-Gal4* (a *Gal4* containing an enhancer from the *AstA* gene) phenocopied the sleep-promoting effects caused by hyperactivating dFB neurons with NaChBac (*Figure 1C,D and E*). The neuronal hyperactivation with NaChBac did not impair locomotion since the levels of activity during the wake periods were similar between the control flies and the animals expressing *UAS-NaChBac* (*Figure 1G*).

Flies that are in a sleep-like state have increased arousal thresholds similar to other animals (*Hendricks et al., 2000*; *Shaw et al., 2000*). Therefore, to provide evidence that the decreased activity due to hyperactivation of *65D05-Gal4*-expressing neurons reflects an increase in sleep, we applied a simple arousal paradigm. We maintained the flies for five days under 12 hr light/12 hr dark cycles. On the fifth night, we exposed the flies to three light-pulses (5 min each delivered at ZT16, ZT18, and ZT20; *Figure 1—figure supplement 1A*). In control flies (*65D05-Gal4/+* and *UAS-*

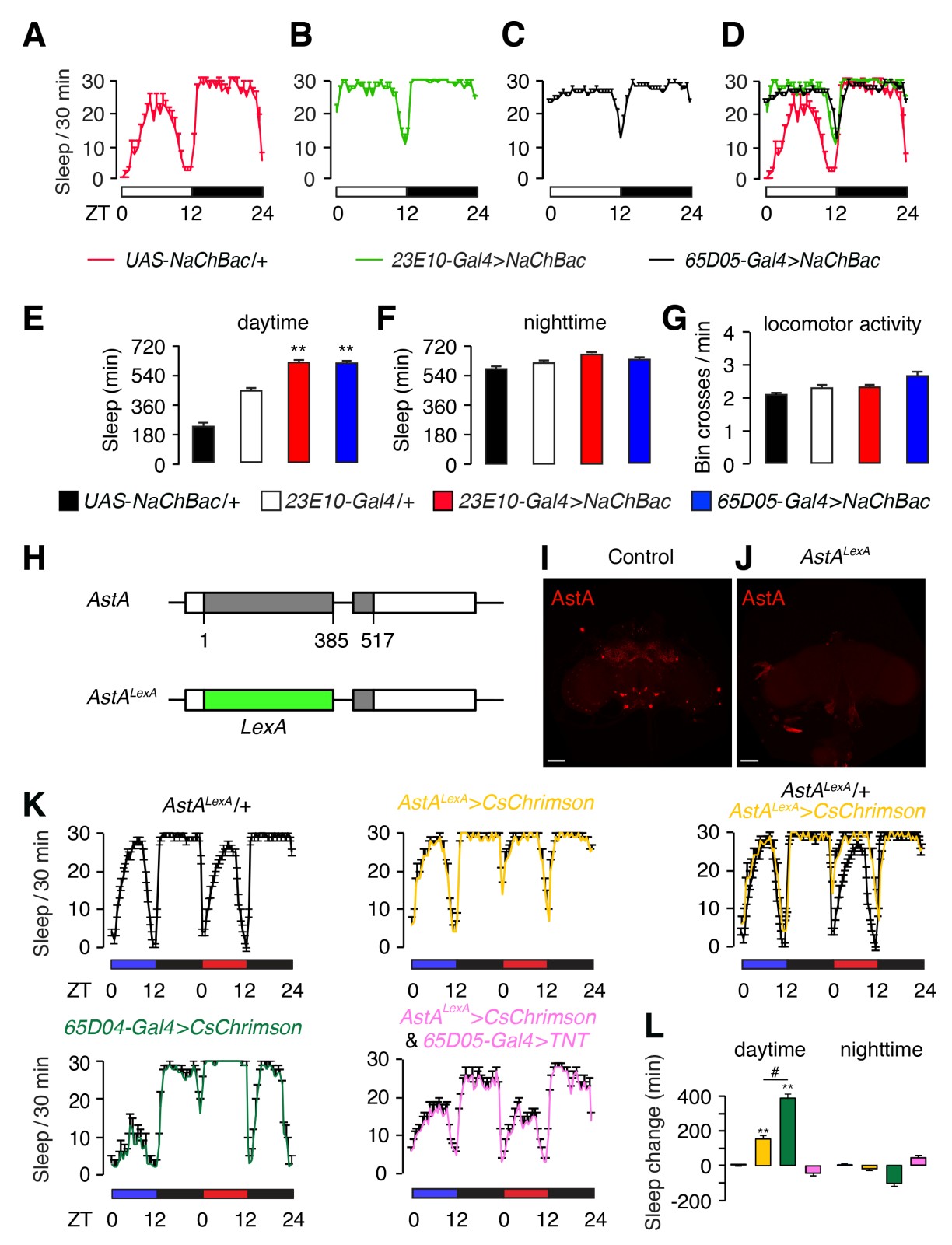

**Figure 1.** Effects on sleep resulting from hyperactivation of dFB (*23E10*) and AstA (*65D05*) neurons. (A—D) Sleep profiles of the indicated flies during the day (ZT0—ZT12) and night (ZT12—ZT24). The white and black bars beneath the sleep profiles indicate the day and night cycles. The amounts of sleep (0—30 min) are plotted per 30 min bins. (**A**) Control flies (*UAS-NaChBac/+*). (**B**) Flies expressing *NaChBac* (*UAS-NaChBac*) using the *23E10-Gal4*. Note that the *23E10-Gal4 > NaChBac* annotation indicates transgenic flies that bear one copy of the *23E10-Gal4* transgene and one copy of the *UAS-*

*Figure 1 continued on next page*

*Figure 1 continued*

*NaChBac* transgene. Similar annotation applies elsewhere as all flies contain one copy each of the *UAS* and *Gal4* transgenes. (C) *65D05-Gal4 > NaChBac*. (D) Combined sleep profiles from A—C. (E,F) Quantification of daytime and nighttime sleep exhibited by the indicated flies. The genotypes are indicated below. (G) Bin crosses/min during the wake periods. The genotypes are indicated below. Error bars, SEMs. \*\*p<0.01, one-way ANOVA with Dunnett's test. n = 16 for *UAS-NaChBac/+*, n = 48 for *23E10-Gal4/+*, n = 35 for *23E10-Gal4 > NaChBac*, and n = 30 for *65D05-Gal4 > NaChBac*. (H) Illustration of the *AstA^LexA* allele containing the *LexA* gene inserted at the position of the normal translation start codon for *AstA*. The numbers indicate the genomic nucleotide residues, with one defined as the first nucleotide of the start codon of the wild-type *AstA* gene. (I,J) Whole-mount *AstA^+* + *AstA^LexA* brains stained with anti-AstA. The scale bars represent 40 µm. (K) Sleep profiles of flies exposed to optogenetic stimulation with CsChrimson. Flies were entrained under 12 hr blue light/12 hr dark cycles for 3 days, and then shifted to a red light/dark cycle on the 4$^{th}$ day. Blue light does not activate CsChrimson. Shown are the sleep profiles under the blue and red light conditions as indicated by the blue and red bars. The total sleep time (0—30 min) is plotted in 30 min bins. The genotypes are indicated below. The flies include one copy of each of the indicated transgenes. *LexOp-CsChrimson/+* and *UAS-CsChrimson/+* were expressed under control of either the *AstA^LexA/+* or *65D04-Gal4/+*, respectively. (L) Quantification of the change in daytime and nighttime sleep induced by CsChrimson activation. *AstA^LexA/+* serves as the control. n = 10–32. Error bars, SEMs. \*\*p < 0.01, one-way ANOVA with Dunnett's test. n = 32 for *AstA^LexA/+*, n = 10 for *AstA^LexA >CsChrimson*, n = 70 for *65D05-Gal4 > CsChrimson*, and n = 23 for *AstA^LexA >CsChrimson* and *65D05-Gal4 > TNT.*.

DOI: https://doi.org/10.7554/eLife.40487.002

The following figure supplements are available for figure 1:

**Figure supplement 1.** Arousal effects of 5 min light delivered at ZT16, ZT18, and ZT20.

DOI: https://doi.org/10.7554/eLife.40487.003

**Figure supplement 2.** Sleep promoting effects resulting from hyperactivation of a subset of *65D05-Gal4* neurons that are not labeled by the *AstA^lexA* reporter.

DOI: https://doi.org/10.7554/eLife.40487.004

*NaChBac/+*), the light stimuli caused large reductions in sleep during the 30-min period following the onset of the light stimuli (*Figure 1—figure supplement 1B—D*). However, the *65D05 > NaChBac* flies did not exhibit decreased sleep (arousal) in response to the 5 min light stimulations (*Figure 1—figure supplement 1B and E*). These data support the proposal that hyperactivation of *65D05-Gal4* positive neurons increases sleep. dFB neurons were proposed to secrete the AstA peptide based on AstA-positive immunostaining around the terminals of dFB neurons (*Donlea et al., 2018*). However, the AstA-expressing neurons are most likely adjacent to rather than the dFB neurons themselves since dFB neurons are not labeled by any of the previously described *AstA* reporters (*Chen et al., 2016*; *Hergarden et al., 2012*). Rather, dFB neurons are labeled by the *23E10-Gal4*, which is an AstA-receptor (AstA-R1) reporter (*Jenett et al., 2012*). Since hyperactivation of neurons labeled by the *AstA* reporter (*65D05-Gal4*) leads to sleep-promoting effects that are similar to those that result from hyperactivation of dFB neurons, which are labeled by the *AstA-R1* reporter, we hypothesize that the neurons that are marked by *AstA* reporter (*65D05-Gal4*) function in the same sleep-promoting pathway as the dFB neurons.

To facilitate the identification of sleep-promoting neurons labeled by the *AstA* reporter, *65D05-Gal4*, we used CRISPR/Cas9-mediated gene editing to replace the coding region with a *LexA* reporter (*AstA^LexA*; *Figure 1H*), so that we could use *LexA* to perform double-labeling with existing *Gal4* reporters. The *AstA^LexA* flies also provided a null allele, which is useful since the currently available line is a hypomorphic mutant (*Donlea et al., 2014*; *Donlea et al., 2018*). As expected, the *AstA^LexA* mutation eliminated anti-AstA staining in the brain (*Figure 1I and J*).

To verify that the *AstA^LexA* reporter indeed labels sleep-promoting neurons, we expressed CsChrimson under control of the *AstA^LexA* reporter (*AstA^LexA/+*) so that *AstA^LexA*-positive neurons could be activated with red lights. We used heterozygous *AstA^LexA/+* flies since *AstA^LexA* is a null allele. Activation of *AstA^LexA*-positive neurons (*AstA^LexA >CsChrimson*) led to an increase in daytime sleep (*Figure 1K and L*). However, the increase was less pronounced than with the *65D05-Gal4* reporter (*Figure 1K and L*).

The *65D05-Gal4* reporter labeled several regions in the brain, including sensory processing regions such as the medulla layer of the optical lobe (med) and the primary gustatory center, the subesophageal zone (SEZ; *Figure 2A and B*), similar to previously characterized *AstA-Gal4* reporter lines (*AstA^1-Gal4* and *AstA^2-Gal4*) (*Hergarden et al., 2012*). However, the *AstA^LexA* reporter labeled many fewer neurons (*Figure 2C*). By comparing the labeling with these two reporters we found that three neurons located in the lateral posterior region of the brain were labeled by both reporters (LPN^AstA; *Figure 2B and C*). The positions of these neurons resemble a group of circadian

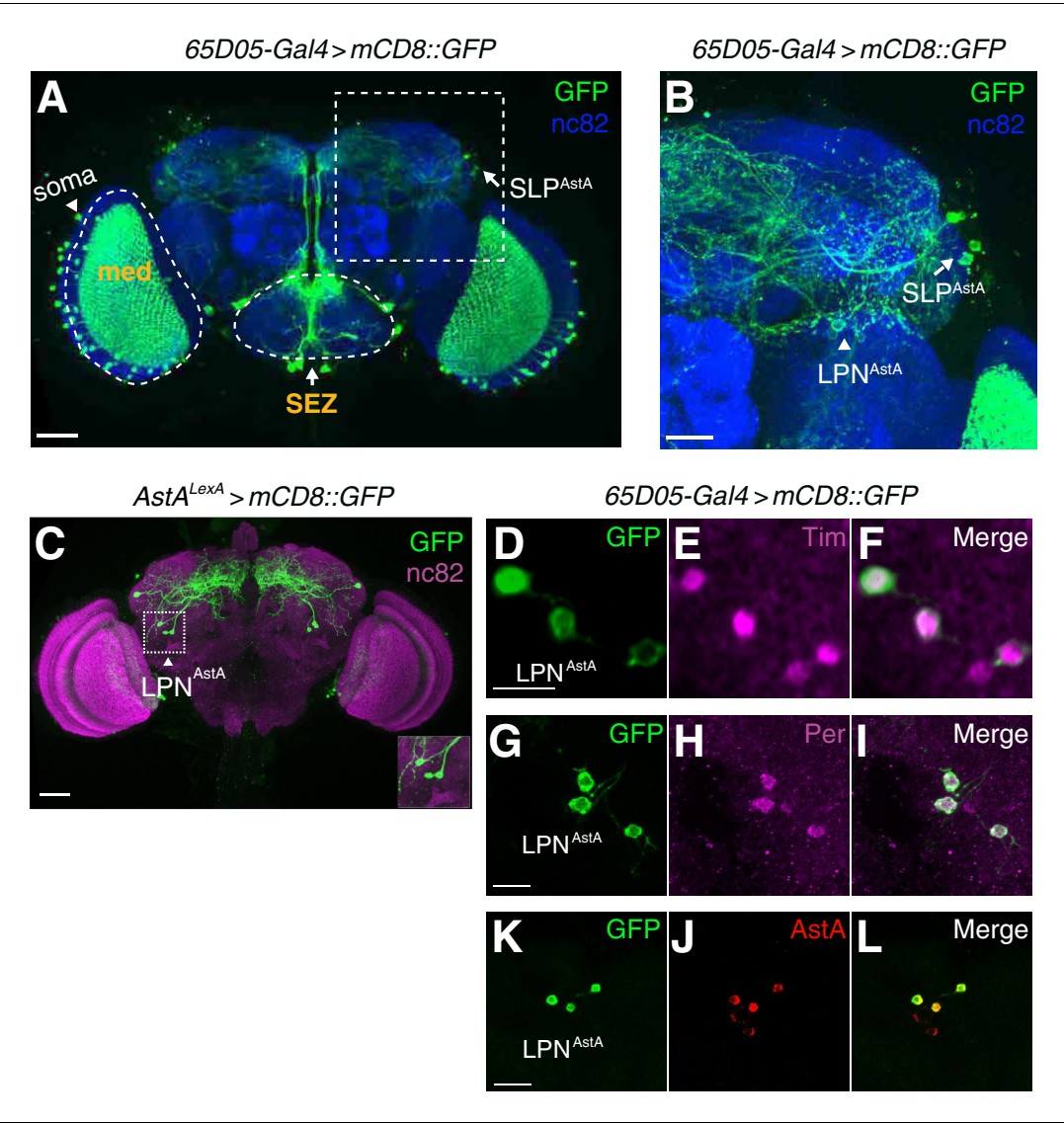

**Figure 2.** LPN^AstA circadian pacemaker neurons are labeled by *AstA* reporters (*65D05-Gal4* and *AstA^LexA*) and anti-AstA. (A) Whole-mount of a brain expressing *UAS-mCD8::GFP* under the control of the *65D05-Gal4*. Green, anti-GFP; blue: anti-nc82 (a pan-neuronal marker labeling active zones). The dashed box indicates the region shown at higher magnification in B. The scale bar represents 40 μm. (B) Zoomed-in view of boxed region in A. The arrow and arrowhead indicate SLP^AstA and LPN^AstA neurons, respectively. The scale bar represents 20 μm. (C) Immunostaining of a brain whole-mount (*AstA^LexA >mCD8::GFP*) with anti-GFP (green) and anti-nc82 (magenta) The scale bar represents 40 μm. The inset at the right bottom corner shows the three LPN^AstA neurons. (D—L) Immunostaining of LPN^AstA neurons in brain whole-mounts from *65D05-Gal4 > mCD8::GFP* flies. (D—F) Co-staining with anti-GFP (green) and anti-Tim (magenta). The scale bar represents 10 μm. (G—I) Co-staining with anti-GFP (green) and anti-Per (magenta). The scale bar represents 10 μm. (J—K) Co-staining with anti-GFP (green) and anti-AstA (red), the scale bar represents 10 μm.

DOI: https://doi.org/10.7554/eLife.40487.005

pacemaker neurons termed lateral posterior neurons (LPNs) (*Shafer et al., 2008*). The contributions of LPNs to sleep have not been reported previously. We confirmed the identity of the LPNs by co-staining with Timeless (Tim; *Figure 2D—F*) and Period (Per; *Figure 2G—I*), two core clock proteins expressed in *Drosophila* circadian pacemaker neurons (*Reddy et al., 1984*; *Sehgal et al., 1994*). These LPNs also express the AstA peptide and are referred to here as LPN^AstA neurons (*Figure 2J—L*).

We tested whether LPN^AstA are the key *AstA^LexA*-positive neurons required for promoting sleep by using tetanus toxin to inhibit neurotransmission from LPN^AstA neurons while activating the remaining *AstA^LexA* neurons. To do so, we took advantage of the observation that the only neurons labeled

by both the *65D05-Gal4* and *AstA^LexA* reporters are LPN^AstA neurons. Therefore, we introduced the *65D05-Gal4 and UAS-tetanus toxin* (*TNT*) (*Sweeney et al., 1995*) into the *AstA^LexA/+ > CsChrimson* background. We treated these animals with red lights and found that the sleep-promoting effect induced in *AstA^LexA/+ > CsChrimson* flies was abolished (*Figure 1K and L*). Based on this evidence we conclude that LPN^AstA neurons are sleep-promoting circadian pacemaker neurons.

Because the increase in sleep caused by light activation of neurons in *AstA^LexA/+ > CsChrimson* flies was less robust than in *65D05-Gal4 > CsChrimson* flies (*Figure 1K and L*), it is plausible that a second group of sleep-promoting neurons are labeled by the *65D05-Gal4* but not the *AstA^LexA* reporter. To test this possibility, we examined the sleep promoting effects resulting from hyperactivation of all neurons labeled by the *65D05-Gal4* except the LPN^AstA neurons. We expressed *UAS-NaChBac* under the control of the *65D05-Gal4*, and in the same flies expressed *Gal80* under the control of *AstA^LexA* to inhibit Gal4 activity in LPN^AstA neurons. We still observed sleep-promoting effects in these transgenic animals including increased total daytime sleep and sleep bout length (*Figure 1—figure supplement 2A—C, E—G*). The sleep-promoting effects displayed by these flies were smaller than without the *Gal80* expression in LPN^AstA neurons (*Figure 1—figure supplement 2D—G*). Together, these findings support the model that LPN^AstA and additional neurons marked by the *65D05-Gal4* reporter promote sleep.

To identify the additional group of sleep-promoting neurons labeled by the *65D05-Gal4*, we used a genetic '*FlpOut*' approach to generate flies expressing the warm-activated cation channel TRPA1 tagged with mCherry in different subgroups of neurons labeled by the *65D05-Gal4* (*Figure 3—figure supplement 1A and B*). To visualize the key sleep-promoting neurons, we compared the expression patterns of *mCherry::trpA1* in the brains of flies that did not show an increase in total sleep with the staining patterns from flies that showed significant increases in sleep upon thermoactivation. Neurons located in the medullar layer (med) of the optic lobe and the SEZ did not show differential expression between the two groups of flies, indicating that they are not the sleep-promoting neurons (*Figure 3—figure supplement 1E—H*). In contrast, we found that neurons located in the superior lateral protocerebrum (SLP) were stained positive by mCherry only in flies exhibiting an increase in sleep (*Figure 3—figure supplement 1I and J*). LPN^AstA neurons stained strongly with mCherry in flies that showed significantly higher levels of sleep, and stained weakly in flies that did not show an increase in sleep (*Figure 3—figure supplement 1I and J*). These data indicate that SLP^AstA neurons are a second group of sleep-promoting neurons labeled by *65D05-Gal4*.

Next, we tested if neuronal activity in the dFB was required for LPN^AstA and SLP^AstA neurons to promote sleep. To do so, we monitored sleep in flies in which we activated LPN^AstA and SLP^AstA neurons and tested the effects of this activation after suppressing neurotransmission from dFB neurons. To stimulate LPN^AstA and SLP^AstA neurons, we expressed CsChrimson (*LexOp-CsChrimson*) using a *65D05-LexA* reporter (constructed with the same DNA regulatory sequences as the *65D05-Gal4*), which also labels LPN^AstA and SLP^AstA neurons similar to the *65D05-Gal4* (*Figure 3—figure supplement 2*). To maintain flies under 12 hr light/12 hr dark cycles without stimulating the red-light activated CsChrimson, we exposed the flies to blue light during the 12 hr light periods for 3 days. On the 4th day, we replaced the blue with red lights to stimulate the neurons. As expected, red light induced an increase in daytime sleep and consolidated daytime sleep into longer bouts (*Figure 3A—F*). However, when we blocked neurotransmission from dFB neurons with TNT (*23E10-Gal4* and *UAS-TNT*), optogenetic activation of LPN^AstA and SLP^AstA neurons no longer promoted sleep or sleep bout consolidation (*Figure 3A—F*). Taken together, we identified two groups of neurons, which promote sleep and appear to function presynaptic to dFB neurons. The finding that one of the groups of neurons are *bona fide* circadian pacemaker neurons suggests a direct neuronal pathway through which the circadian system regulates the sleep homeostat.

## SLP^AstA and LPN^AstA neurons form close associations and function presynaptic to dFB neurons

To address whether SLP^AstA and LPN^AstA neurons form close associations with dFB neurons, we performed double-labeling experiments. We examined the relative positions of the projections of SLP^AstA, LPN^AstA and dFB neurons by expressing different fluorescent reporters. Both SLP^AstA and LPN^AstA neurons send their projections to the superior median protocerebrum (SMP) region, which is also innervated by projections from dFB neurons (*Figure 3G and H*), indicating the potential for a direct neuronal connectivity.

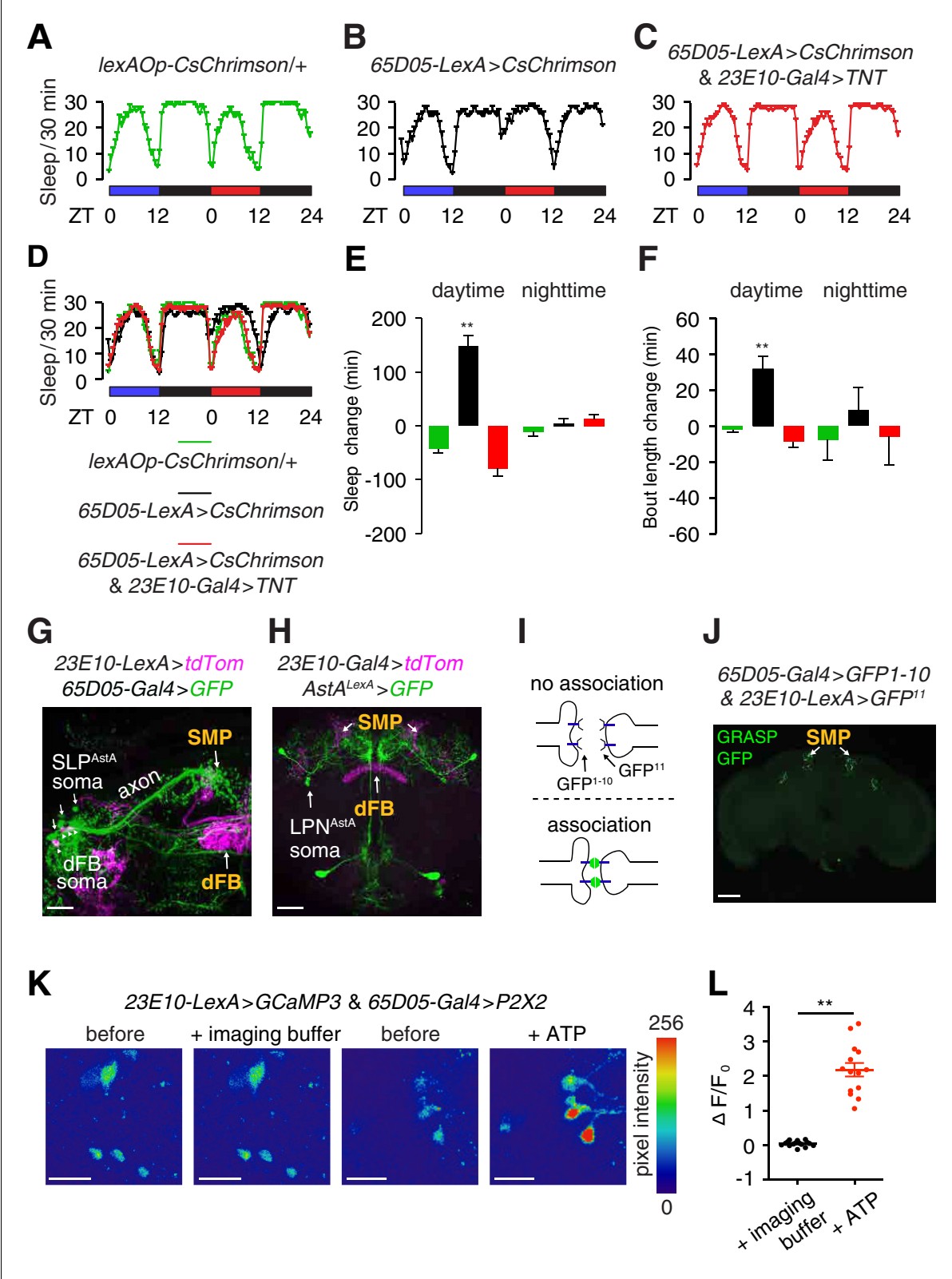

**Figure 3.** Anatomical and functional connectivity between LPN[AstA]/SLP[AstA] and dFB neurons. (A—D) Sleep profiles of the indicated flies under 12 hr blue light/12 hr dark and 12 hr red light/12 hr dark cycles. Red but not blue lights activate CsChrimson. (E) Quantification of changes in daytime and nighttime sleep due to neuronal activation by red lights in *65D05* neurons expressing CsChrimson. (F) Quantification of changes in daytime and nighttime sleep bout length due to neuronal activation by red lights in *65D05* neurons expressing CsChrimson. The changes in total sleep time and the

*Figure 3 continued*

average bout lengths were calculated by subtracting these sleep parameter values obtained during the blue-light/dark cycles from those obtained during the red-light/dark cycles. Error bars, SEMs. **p < 0.01, one-way ANOVA with Dunnett's test. n = 15 for *lexAOp-CsChrimson/+*, n = 24 for *65D05-LexA > CsChrimson*, and n = 31 for *65D05-LexA > CsChrimson* and *23E10-Gal4 > TNT*. (G) Anti-tdTomato (magenta) and anti-GFP (green) staining of a whole-mount brain expressing tdTomato in dFB neurons (*23E10-LexA > tdTom*) and GFP in SLP[AstA] neurons (*65D05-Gal4 > GFP*). Both reporters showed innervation in the SMP. The scale bar represents 20 μm. (H) Anti-tdTomato (magenta) and anti-GFP (green) staining of a whole-mount brain expressing tdTomato in dFB neurons and GFP in LPN[AstA] neurons (*AstA[LexA] >GFP*). Both reporters stained the SMP region. The scale bar represents 40 μm. (I) Cartoon illustrating the GRASP assay. Fluorescence is produced only when the two segments of GFP associate on the extracellular surfaces of adjacent cells. (J) Image of GRASP GFP fluorescence revealing close association between *65D05-Gal4* and *AstAR1-LexA* labeled neurons in the brain. The scale bar represents 60 μm. (K) Representative images of GCaMP3 fluorescence in dFB neurons upon activating the ATP-gated P2X2 cation channel in LPN[AstA] and SLP[AstA] neurons (*65D05-Gal4 > P2X2*) Shown are images before and during application of the imaging buffer (AHL) only, or ATP (2.5 mM final concentration) in imaging buffer. The scale bars represent 20 μm. (L) Quantification of the changes in GCaMP3 fluorescence in dFB neurons before and after adding the imaging buffer only or during exposure to 2.5 mM ATP to activate P2X2 expressing neurons. The effects of adding ATP versus the imaging buffer only were compared. Error bars, SEM. **p<0.01, unpaired Student's *t*-test. n = 14 for ATP treatment and 16 for treatment with the imaging buffer from three independent imaging experiments.

DOI: https://doi.org/10.7554/eLife.40487.006

The following figure supplements are available for figure 3:

**Figure supplement 1.** LPN[AstA] and SLP[AstA] are the sleep-promoting neurons labeled by the *65D05-Gal4*.

DOI: https://doi.org/10.7554/eLife.40487.007

**Figure supplement 2.** Expression pattern of *65D05-LexA* and *65D05-Gal4*.

DOI: https://doi.org/10.7554/eLife.40487.008

We then used GFP reconstitution across synaptic partners (GRASP) (*Feinberg et al., 2008*; *Gordon and Scott, 2009*) to test if these two sets of sleep-promoting neurons form close associations. In this approach two non-functional fragments of membrane tethered GFP (mCD4-GFP[1-10] and mCD4-GFP[11]) are expressed on the extracellular surfaces of different neurons. GFP fluorescence is reconstituted only if the two fragments are brought into contact, such as when the terminals of the two neurons associate closely (*Figure 3I*). Consistent with our double-labeling experiments (*Figure 3G and H*), we detected GFP fluorescence in the SMP region of the brain where LPN[AstA], SLP[AstA] (expressing *65D05-Gal4 > UAS-mCD4-GFP[1-10]*) and dFB neurons (expressing *23E10-LexA > lexAOp-mCD4-GFP[11]*) send their projections (*Figure 3J*, *Figure 3—figure supplement 1K and L*).

To investigate functional connectivity, we expressed the ATP-activated cation channel P2X2 (*UAS-P2X2*) (*Yao et al., 2012*) in SLP[AstA] and LPN[AstA] neurons under control of the *65D05-Gal4*, and monitored changes in the activity of dFB neurons with GCaMP3 (*UAS-GCaMP3*), a genetically encoded Ca$^{2+}$ indicator (*Tian et al., 2009*). After applying ATP to activate the SLP[AstA] and LPN[AstA] neurons, there was a significant increase in GCaMP3 fluorescence in dFB neurons relative to applying the imaging buffer only (*Figure 3K and L*).

## Glutamate is a sleep-promoting neurotransmitter used by LPN[AstA] and SLP[AstA] to activate dFB neurons

Based on the results described above we propose the existence of sleep-promoting excitatory synaptic connections from LPN[AstA] and SLP[AstA] neurons to the downstream dFB neurons. To identify the relevant neurotransmitter, we used RNAi to knock-down genes essential for the synthesis or packaging of various neurotransmitters. These include the vesicular glutamate transporter (*VGlut*) (*Daniels et al., 2006*), the vesicular monoamine transporter (*VMAT*) (*Greer et al., 2005*), choline acetyltransferase (*ChAT*) (*Itoh et al., 1986*) and glutamic acid decarboxylase 1(*Gad1*) (*Jackson et al., 1990*).

To screen for RNAi lines that lead to a large reduction of target gene expression, we expressed a panel of available RNAi transgenes targeting the above genes with the pan-neuronal reporter *elav-Gal4*. We used a *VGlut-RNAi* line and a *VMAT-RNAi* line that lead to substantial reductions of target gene expression (*Figure 4—figure supplement 1A—F* and Methods). For *ChAT* and *Gad1* we selected the RNAi lines that lead to lethality when expressed pan-neuronally, which is indicative of effective abolition of acetylcholine and GABA synthesis since they are essential neurotransmitters.

We first expressed these RNAi transgenes in flies expressing *UAS-NaChBac* under control of the *65D05-Gal4*, which results in hyperactivation of *AstA*-positive neurons, including LPN[AstA] and SLP[AstA] neurons (*65D05 > NaChBac* and RNAi). We found that the increase in sleep due to hyperactivation by NaChBac (*65D05 > NaChBac*) was reduced when RNAi was directed against *VGlut*—the transporter required for packaging glutamate in synaptic vesicles (*Figure 4A and F*). In contrast, the elevation in sleep induced by NaChBac was not affected when we used RNAi to knockdown genes required for the synthesis or packaging of other major neurotransmitters including acetylcholine (*ChAT*), γ-aminobutyric acid (GABA; *Gad1*) and biogenic amines (*VMAT*) (*Figure 4B—F*). These observations suggest an important role for glutamate in conferring the sleep-promoting contributions of LPN[AstA] and SLP[AstA] neurons.

To investigate the role of glutamate in basal sleep, we expressed the *VGlut-RNAi* without neuronal hyperactivation (no NaChBac expression). While overall sleep time was unaffected (*Figure 4—figure supplement 1G and H*), knocking down *VGlut* in *65D05-Gal4* positive neurons increased fragmented nighttime sleep as indicated by a significant reduction of nighttime sleep bout length (*Figure 4—figure supplement 1J*). The small reduction in daytime sleep bout length was not statistically significant (*Figure 4—figure supplement 1I*). A greater reduction of sleep bout length (both day and night) occurred when we hyperpolarized (*65D05 > kir2.1*) or blocked neurotransmission (*65D05 > TNT*) from SLP[AstA] and LPN[AstA] neurons (*Figure 4—figure supplement 1K and L*).

To test whether the dFB neurons respond to glutamate, we employed an ex-vivo brain preparation amenable to imaging neuronal activity in freshly dissected brains. We expressed a fluorescent Ca$^{2+}$ sensor (*UAS-GCaMP6f*) (*Chen et al., 2013*) to image the activities of dFB neurons after applying glutamate to the bath. We found that addition of glutamate caused an increase in GCaMP6f fluorescence (*Figure 4G and H*). In contrast, these neurons did not respond to acetylcholine—another major excitatory neurotransmitter in the *Drosophila* brain (*Figure 4I and J*).

## Sleep-promoting role of AstA and a central clock component in LPN[AstA] neurons

LPN[AstA] neurons express both the neuropeptide AstA and central clock genes (*Figure 2D—L*). To test for a potential contribution of the central clock for the sleep-promoting function of LPN[AstA] neurons we expressed a dominate-negative isoform of the clock component, Clock (*UAS-ClkΔ*) (*Tanoue et al., 2004*) in LPN[AstA] neurons under control of the *65D05-Gal4*. We found that the sleep-promoting effect of neuronal hyperactivation of LPN[AstA] neurons (*65D05-Gal4 > NaChBac*) during the daytime was reduced by expression of *ClkΔ* (*Figure 4—figure supplement 2A and B*). This indicates that the central clock in LPN[AstA] neurons contributes to the sleep promoting function of these neurons.

To address whether AstA functions in LPN[AstA] neurons to promote sleep, we examined whether removal of AstA impacted on the sleep-promoting effect induced by hyperactivation of these neurons (*65D05-Gal4 > NaChBac*). We found that the *AstA[LexA]* mutation significantly decreased the sleep-promoting effect caused by expression of NaChBac (*Figure 4—figure supplement 2C and D*). The reduction in sleep due to the *AstA[LexA]* mutation was most pronounced during the second half of the daytime period. These results indicate that in addition to glutamate (*Figure 4A*) AstA also contributes to the sleep-promoting function of LPN[AstA] neurons.

## Synaptic inputs from sleep- and arousal-promoting neurons target different dFB projections

Dopamine is released by tyrosine hydroxylase (TH)-expressing neurons and is an arousal-promoting molecule (*Liu et al., 2012*; *Pimentel et al., 2016*; *Ueno et al., 2012b*), which inhibits the activity of dFB neurons (*Figure 5—figure supplement 1A and B*). We imaged the relative innervation patterns of dopamine arousal (DAA) neurons and dFB neurons to address whether they were in close association. We labeled dFB neurons with tdTomato (*23E10-LexA > tdTomato*; *Figure 5A* and *B*), and DAA neurons with a GFP reporter (*TH-Gal4 >GFP*, *Figure 5A and B*), which was expressed in a majority of neurons that stained with anti-TH (*Figure 5—figure supplement 1C—E*). The soma of the DAA and dFB neurons were distinct, although many were juxtaposed (*Figure 5—figure supplement 1F*). We found that the DAA neurons project to two layers within the FB: the dorsal FB layer (dFB) and the ventral FB layer (vFB; *Figure 5B*). The dorsal layer innervation of DAA neurons overlapped with

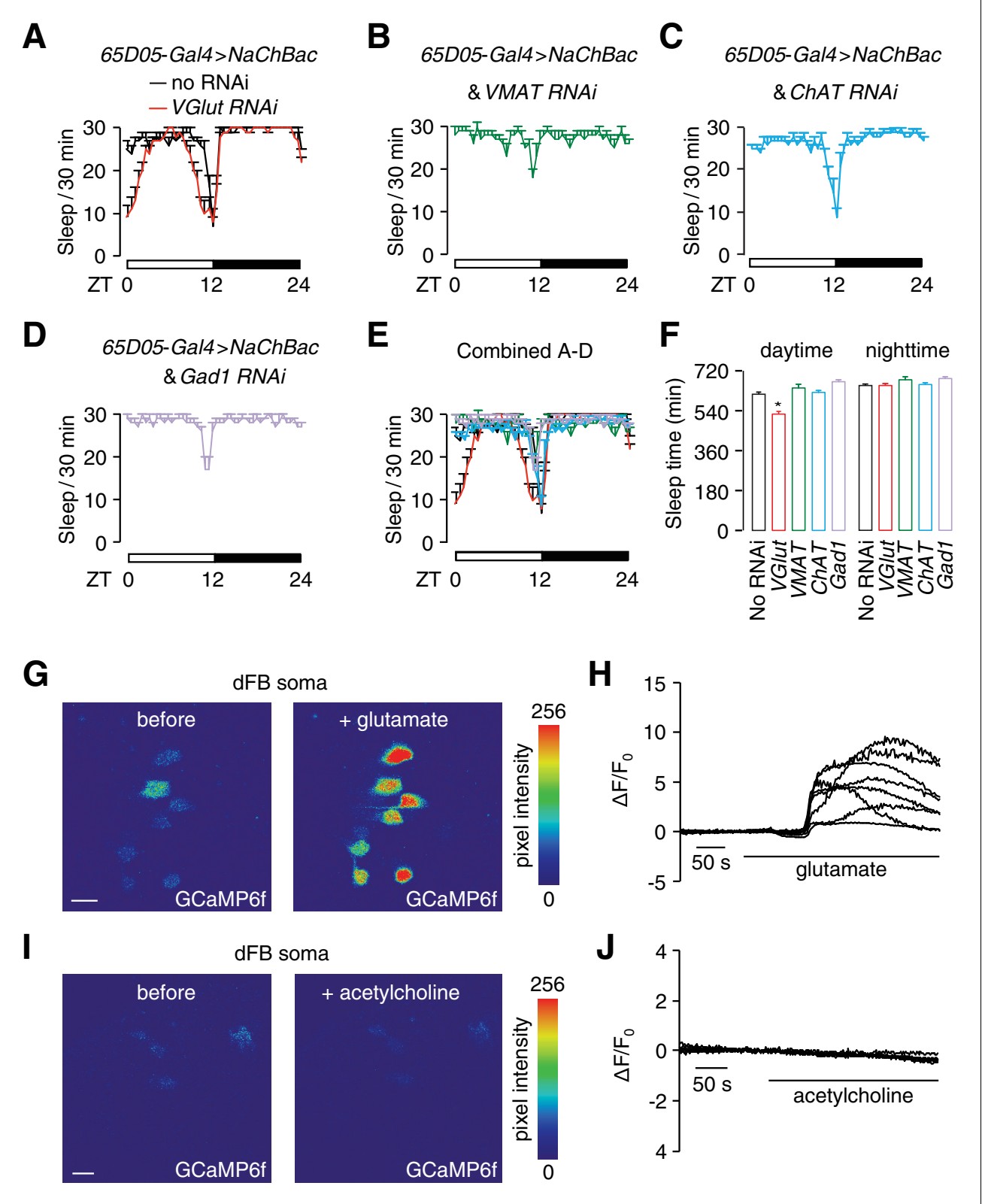

**Figure 4.** Glutamate is the excitatory neurotransmitter in *65D05-Gal4*-positive neurons, which activates dFB neurons and promotes sleep. (A) Sleep profile of *65D05-Gal4 > NaChBac* flies with and without the *VGlut RNAi* (*UAS-VGlut*) expressed under control of the *65D05-Gal4*. (B—D) Sleep profiles of *65D05-Gal4 > NaChBac* flies expressing *VMAT-RNAi*, *ChAT-RNAi* or *Gad1-RNAi* transgenes under control of the *65D05-Gal4*. (E) Combination of sleep profiles shown in A—D. (F) Quantification of daytime and nighttime sleep of *65D05-Gal4 > NaChBac* flies expressing the indicated RNAi lines.

*Figure 4 continued on next page*

*Figure 4 continued*

Error bars, SEM. **p<0.01, one way ANOVA with Dunnett's test. n = 85 for no RNAi, n = 51 for *VGlut* RNAi, n = 42 for *VMAT RNAi*, n = 46 for *ChAT RNAi*, and n = 40 for *Gad1 RNAi*. (**G**) Representative images of GCaMP6f fluorescence before and after bath application of 50 mM glutamate. (**H**) Representative traces of changes in GCaMP6f fluorescence ($\Delta F/F_0$) upon bath application of 50 mM glutamate. (**I**) Representative images of GCaMP6f fluorescence before and after bath application of 50 mM acetylcholine. (**J**) Representative traces showing changes in GCaMP6f fluorescence ($\Delta F/F_0$) upon bath application of 50 mM acetylcholine. The scale bars in panels **G** and **I** represent 20 μm.

DOI: https://doi.org/10.7554/eLife.40487.009

The following figure supplements are available for figure 4:

**Figure supplement 1.** RNAi knock-down of *VGlut* and *VMAT*.

DOI: https://doi.org/10.7554/eLife.40487.010

**Figure supplement 2.** Testing whether *Clk* and *AstA* function in sleep in LPN[AstA] neurons.

DOI: https://doi.org/10.7554/eLife.40487.011

the projections from dFB neurons (*Figure 5B*), suggesting possible synaptic connection between DAA and dFB neurons.

We used GRASP to assess whether the DAA and dFB neuronal membranes were in close proximity, which would support a model that they make synaptic connections. To test this possibility, we imaged brains from flies expressing two fragments of GFP (*UAS-CD4::spGFP[1-10]* and *lexAOp-CD4::spGFP[11]*) under the control of the *TH-Gal4* and the *23E10-LexA* (*Figure 5—figure supplement 1G*). We observed GRASP signals in regions of the brain containing dFB neuronal projections, including the SMP region and the dFB (*Figure 5C*). We did not detect GFP signals in brains expressing only one of the two GFP fragments (*Figure 3—figure supplement 1L* and *Figure 5—figure supplement 1H*).

To determine if activation of DAA neurons inhibits dFB neurons, we expressed the ATP-activated cation channel P2X2 in DAA neurons (*TH-Gal4 >P2X2*) and monitored changes in the activity of dFB neurons with GCaMP3 (*23E10-LexA > GCaMP3*). After applying ATP, we observed a significant reduction (p < 0.01, unpaired Student's *t*-test) in GCaMP3 fluorescence relative to the effect of applying the imaging buffer only (*Figure 5D and E*).

While LPN[AstA] and SLP[AstA] sleep promoting neurons only synapse onto the SMP projections of dFB neurons, DAA arousal neurons synapse onto both the SMP projections and the dorsal layer projections of dFB neurons (*Figure 5F*). To characterize these synaptic inputs further, we imaged the relative dendritic and axonal locations of dFB neurons by expressing both a dendritic marker (*UAS-DenMark*) and an axonal marker (*UAS-syt::eGFP*) (*Nicolaï et al., 2010*) under control of the dFB reporter (*23E10-Gal4*). We found that SMP projections of dFB neurons were more intensively marked by DenMark, while the dorsal layer projections of dFB neurons were more heavily labeled with syt::eGFP (*Figure 5G and H*). Therefore, we propose that dFB neurons receive upstream inputs mainly through synapses in the SMP area stemming from sleep-promoting LPN[AstA] and SLP[AstA] neurons. Furthermore, these data suggest that the dFB neurons transmit signals through axonal terminals located in the dorsal FB layer.

One possible role of the synaptic connections from DAA neurons to the axonal terminals of dFB neurons is pre-synaptic regulation of dFB neuronal output. Therefore, we asked if dopamine suppresses $Ca^{2+}$ dynamics in axonal terminals of dFB neurons located in the dorsal layer, thereby negatively regulating the output from dFB neurons. To test this idea, we imaged $Ca^{2+}$ dynamics locally in the axonal terminals of dFB neurons. To do so, we expressed syt::GCaMP6s (*Cohn et al., 2015*) (a protein consisting of synaptotagmin fused to GCaMP6s) in dFB neurons and performed imaging using an ex-vivo brain preparation. When we applied dopamine to the bath, we observed a reduction in $Ca^{2+}$ levels in the axonal terminals of the dFB neurons (*Figure 5I and J*). In contrast, application of either of two other major invertebrate biogenic amines (octopamine and tyramine) did not change the $Ca^{2+}$ dynamics in dFB neurons (*Figure 5—figure supplement 1I*).

## Effect of simultaneous activation of dFB and DAA neurons

Since the activity of dFB neurons are regulated by LPN[AstA]/SLP[AstA] neurons and DAA neurons in opposite directions, we tested the effects on sleep when these two groups of dFB regulatory neurons (LPN[AstA]/SLP[AstA] and DAA neurons) were activated simultaneously. Activating DAA neurons (*TH-Gal4 >NaChBac*) led to a drastic reduction in total sleep time (158 ± 37 min), relative to the

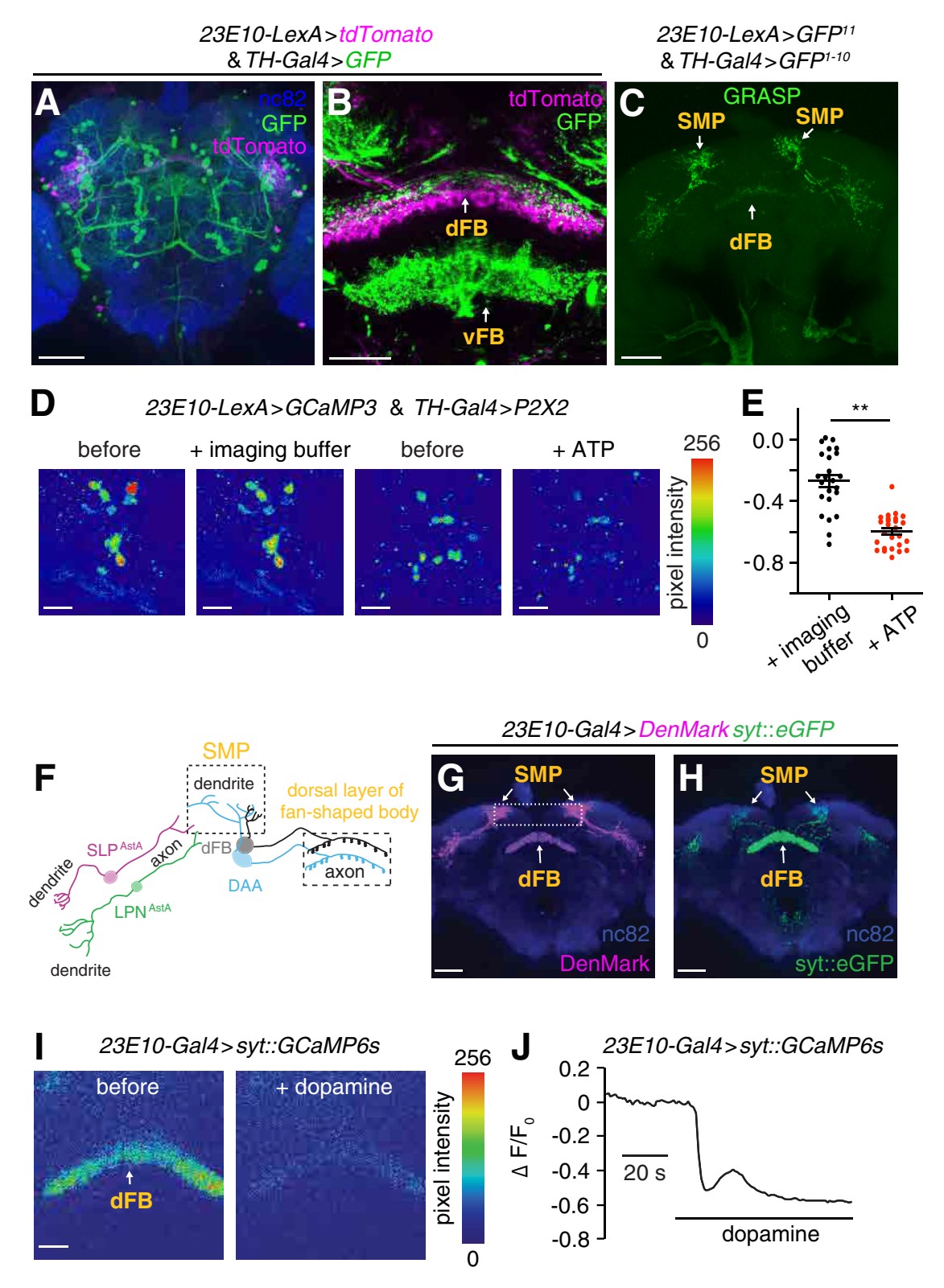

**Figure 5.** Dopaminergic arousal (DAA) neurons associate closely with dFB neurons and downregulate their activity. (**A, B**) Brains expressing GFP in dopaminergic neurons (*TH-Gal4 >GFP*) and tdTomato in dFB neurons (*23E10-LexA > tdTomato*) stained with anti-GFP (green), anti-tdTomato (magenta) and anti-nc82. The dFB and vFB regions are indicated in B. The scale bars represent 60 and 20 μm in A and B, respectively. (**C**) GRASP GFP fluorescence revealing close associations between sleep-promoting dFB neurons and dopaminergic neurons in the brain. The dFB (*23E10-LexA*

*Figure 5 continued on next page*

*Figure 5 continued*

positive) and dopaminergic neurons (*TH-Gal4* positive) expressed the GFP[11] and GFP[1-10] fragments, respectively. The scale bar represents 60 µm. (**D**) Representative images of GCaMP3 fluorescence in dFB neurons (*23E10-LexA* positive) upon activating the ATP-gated P2X2 cation channel (*UAS-P2X2*) expressed in DAA neurons under control of the *TH-Gal4*. Shown are images before and during application of the imaging buffer (AHL) only, or ATP (2.5 mM final concentration) in the imaging buffer. The scale bars represent 20 µm. (**E**) Quantification of the changes in GCaMP3 fluorescence in dFB neurons before and after adding the imaging buffer (AHL) only or during exposure to 2.5 mM ATP to activate P2X2 expressing DAA neurons. The genotype of the flies is as indicated in (**D**). \*\*p < 0.01. Unpaired Student's *t*-test. n = 24 for ATP treatment and n = 26 for treatment with the imaging buffer only from three independent imaging experiments. (**F**) Cartoon showing the positions of the dendrites and axons of LPN[AstA]/SLP[AstA] neurons and DAA neurons relative to the processes of the dFB neurons. (**G, H**) Whole-mount brain expressing dendritic (*UAS-DenMark*) and axonal (*UAS-syt-eGFP*) markers in dFB neurons under control of the *23E10-Gal4*. The brain was stained with anti-dsRed and anti-GFP to detect DenMark and syt::eGFP, respectively, and with anti-nc82. The boxed region in (**G**) indicates the superior medial protocerebrum (SMP). The scale bars represent 40 µm. (**I**) Representative images showing syt::GCaMP6s fluorescence in axonal terminals of dFB neurons before the after bath application of 10 mM dopamine. *UAS-syt::GCaMP6s* was expressed in dFB neurons under control of the *23E10-Gal4*. The scale bar represents 20 µm. (**J**) Representative trace showing the change in syt::GCaMP fluorescence ($\Delta F/F_0$) in axons of dFB neurons (*23E10-Gal4 > syt::GCaMP6s*) upon bath application of 10 mM dopamine.
DOI: https://doi.org/10.7554/eLife.40487.012

The following figure supplement is available for figure 5:

**Figure supplement 1.** DAA arousal neurons regulate dFB neurons.
DOI: https://doi.org/10.7554/eLife.40487.013

controls (849 ± 32 min for *UAS-NaChBac*/+and 982 ± 38 for*TH-Gal4*/+; *Figure 6A and B*). As described above, activating *65D05-Gal4* positive neurons (LPN[AstA] and SLP[AstA]) promotes sleep (1343 ± 36 min; *Figure 6A and B*). Of significance, co-activating DAA and LPN[AstA]/SLP[AstA] neurons with NaChBac led to a level of total sleep time (881 ± 46 min) similar to the control flies (*Figure 6A and B*).

To characterize the consequences of co-activating DAA and LPN[AstA]/SLP[AstA] neurons in greater detail, we analyzed two features of sleep patterns: sleep bout number and sleep bout length. When we only used the *65D05-Gal4* to express *NaChBac*, daytime sleep bouts were significantly lengthened (*Figure 6C*). Conversely, activating DAA neurons (*TH >NaChBac*) resulted in fragmented sleep with a significantly reduced sleep bout length, which was most obvious for nighttime sleep (*Figure 6C*). When we simultaneously activated LPN[AstA]/SLP[AstA] and DAA neurons with the *65D05-Gal4* and *TH-Gal4*, respectively, the sleep bouts were shorter than those exhibited by the control but were comparable to flies in which only the DAA neurons were activated (*TH >NaChBac*; *Figure 6C*). Additionally, activating both LPN[AstA]/SLP[AstA] and DAA neurons caused the flies to initiate many more sleep episodes, as indicated by a significant increase in the sleep bout number (*Figure 6D*).

We then compared the episodes of wakefulness of these flies. In control flies, wake bout length is much longer during the day than night (*Figure 6E*). Activating sleep-promoting LPN[AstA]/SLP[AstA] neurons (*65D05-Gal4 > NaChBac*) led to a reduction in the bout length of daytime wakefulness, while activating arousal-promoting DAA neurons increased the bout length of nighttime wakefulness (*Figure 6E*). Remarkably, when the two groups of neurons were activated simultaneously, both daytime and nighttime activities were fragmented (*Figure 6E*).

## GABA production is required in dFB neurons to promote sleep

To screen for the neurotransmitter synthesized in dFB neurons that is essential for conveying sleep-promoting signals, we silenced expression of genes required for neurotransmitter synthesis or packaging. Thermoactivation of dFB neurons with TRPA1 (*23E10 > trpA1*) enhances sleep (*Figure 7A, B, D—F*). However, this effect was eliminated when we used RNAi to knockdown *Gad1* in dFB neurons (*23E10 > trpA1* and *Gad1* RNAi; *Figure 7C, D and G*). In contrast, we still observed significant sleep-promoting effects resulting from TRPA1-induced activation of dFB neurons when we used RNAi to suppress production of other major neurotransmitters (*Figure 7H—J*). These results suggest that GABA is a necessary sleep-enhancing neurotransmitter synthesized in dFB neurons. This observation is in consistent with a previous report that dFB neurons communicate through inhibitory neurotransmitters and express the vesicular GABA transporter (*Donlea et al., 2018*).

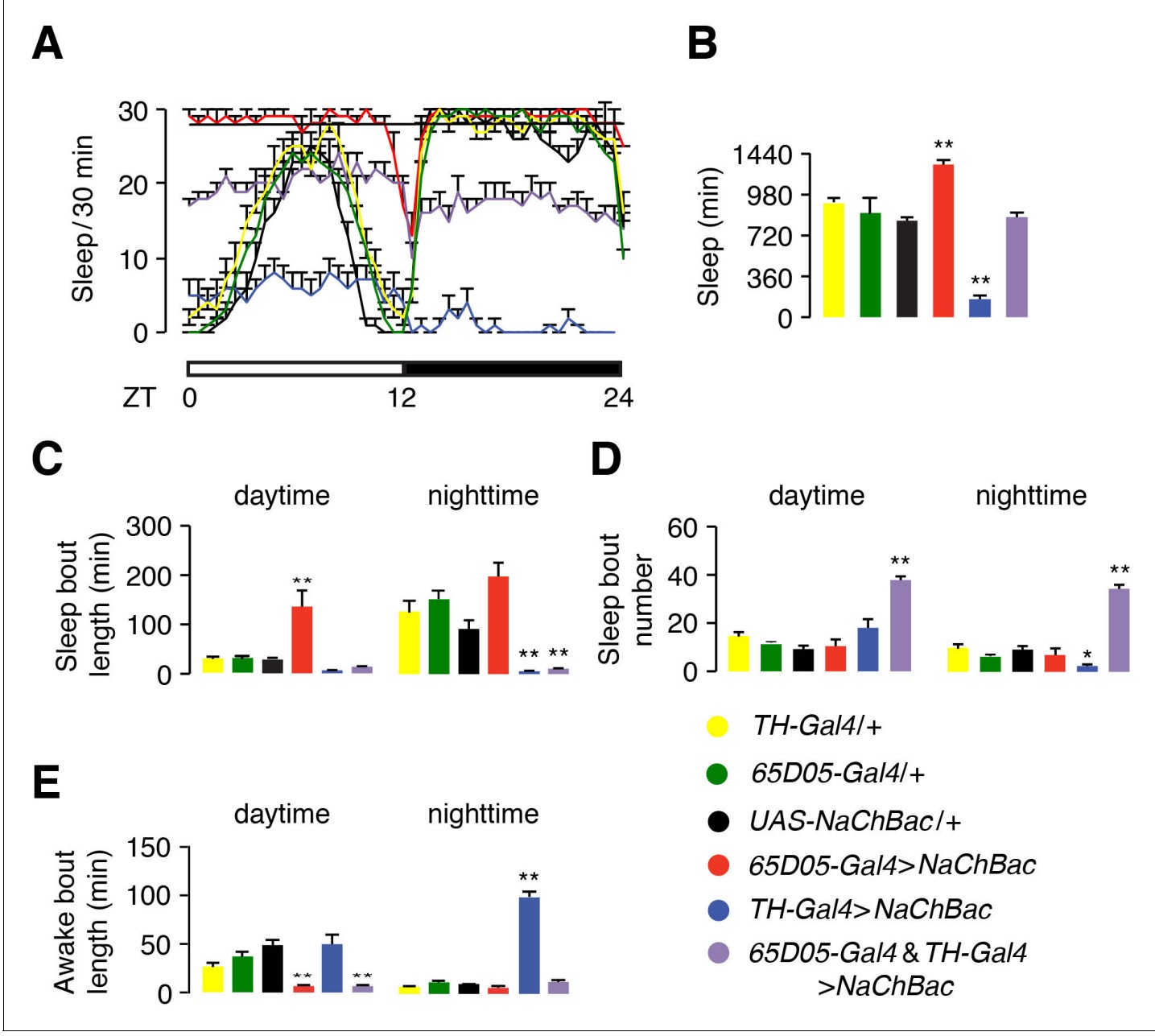

**Figure 6.** Hyperactivation of dopaminergic neurons antagonizes the sleep-promoting effect of hyperactivation of LPN[AstA] and SLP[AstA] (*65D05-positive*) neurons. (A) Effects on sleep profiles due to hyperactivation of *TH-positive* (dopaminergic) and *65D05-positive* (includes LPN[AstA] and SLP[AstA] neurons) with NaChBac. See the legend at the bottom right for the genotypes. (B) Quantification of total sleep time during a 24 hr light/dark cycle by the flies indicated in (A). (C—E) Quantification of daytime and nighttime sleep and awake parameters in flies of the indicated genotypes. (C) Sleep bout length. (D) Sleep bout number. (E) Awake bout length. n = 12—24. Error bars, SEMs. *p < 0.05, **p < 0.01. One-way ANOVA with Dunnett's test. n = 16 for *TH-Gal4/+*, n = 47 for *65D05-Gal4/+*, n = 16 for *UAS-NaChBac/+*, n = 12 for *65D05-Gal4 > NaChBac*, n = 14 for *TH-Gal4 >NaChBac*, and n = 24 for *65D05-Gal4* and *TH-Gal4 >NaChBac*.

DOI: https://doi.org/10.7554/eLife.40487.014

## dFB sleep-promoting neurons inhibit arousal-promoting neurons

Octopamine (OA) appears to increase wakefulness as nighttime sleep is reduced upon feeding flies OA (*Crocker and Sehgal, 2008*) or by activating tyrosine decarboxylase 2 (TDC2)-expressing neurons with NaChBac (*tdc2-Gal4 > NaChBac*) (*Crocker et al., 2010*) (*Figure 8A*). Nighttime sleep in

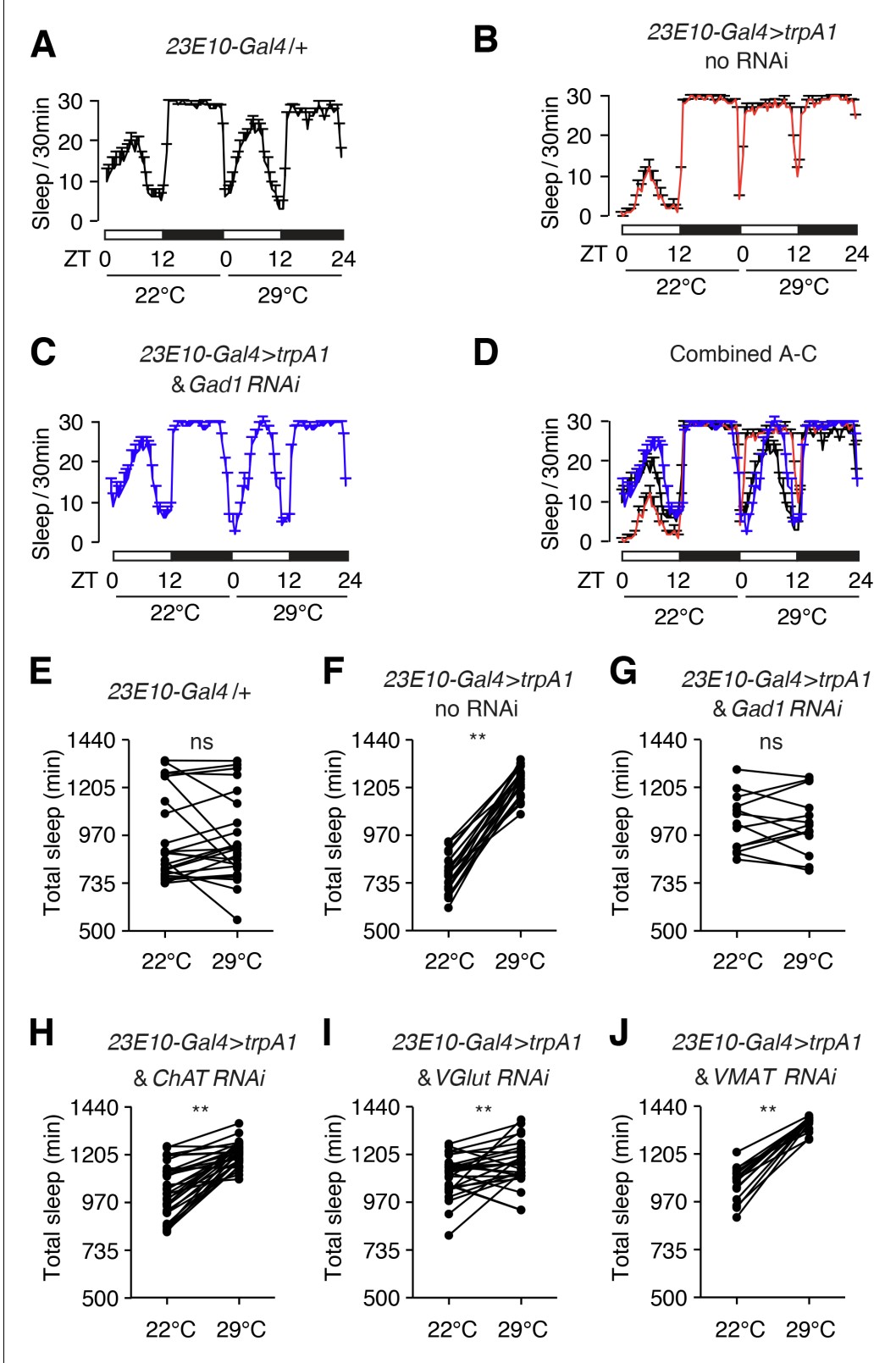

**Figure 7.** Sleep promotion due to hyperactivation of dFB neurons (*23E10-Gal4* positive) depends on GABA produced by dFB neurons. (**A—D**) Elevated sleep due to thermal hyperactivation of dFB neurons with TRPA1 (*23E10-Gal4 > trpA1*) is reduced by expression of the *Gad1* RNAi in dFB neurons. TRPA1 is activated at 29°C but not 22°C. (**E—J**) Quantification of total sleep-time at 22°C and 29°C exhibited by *23E10-Gal4 > trpA1* flies with or without RNAi transgenes directed at genes required for neurotransmitter synthesis or packaging. **p < 0.01. Paired Student's *t*-test. n = 24 for *23E10-Gal4/+*,

*Figure 7 continued*

n = 24 for *23E10-Gal4 > trpA1* no RNAi, n = 13 for *23E10-Gal4 > trpA1* and *Gad1* RNAi, n = 32 for *23E10-Gal4 > trpA1* and *ChAT* RNAi, n = 28 for *23E10-Gal4 > trpA1* and *VGlut* RNAi, and n = 9 for *23E10-Gal4 > trpA1* and *VMAT* RNAi.

DOI: https://doi.org/10.7554/eLife.40487.015

*tdc2-Gal4 > NaChBac* flies is fragmented as the number of sleep bouts increases dramatically, and the average length of each nighttime sleep bout is reduced greatly (*Figure 8B and C*).

We wondered whether dFB neurons might downregulate the activity of TDC2-positive arousal (OAA) neurons. We used two approaches to test for possible close associations between these two sets of neurons. First, we performed double-labeling experiments. The *tdc2-Gal4* reporter labeled neurons that project to multiple layers of the FB, including the dorsal layer where dFB neurons also project (*Figure 8D*). We then expressed dendritic (*UAS-DenMark*) and axonal (*UAS-syt::eGFP*) markers under control of the *tdc2-Gal4* to visualize potential input and output sites of the *Tdc2*-positive (OA) neurons. Most of the OA neuronal projections in the dFB were axonal (syt::eGFP; *Figure 8E*). In addition, there was a single dendritic layer in the dFB labeled with DenMark (*Figure 8E*). To determine whether dFB and OA neurons were in close contact, we performed GRASP analysis and observed signals in multiple areas of the brain. Of note, the OA neurons, which express the *Tdc2-Gal4*, form close connections with the dFB axonal terminals, which express *23E10* (*Figure 8F*).

To determine if activation of dFB neurons inhibited octopaminergic arousal (OAA) neurons, we expressed the ATP-activated cation channel P2X2 in dFB neurons and imaged OAA neuronal activity with GCaMP3. After applying ATP we observed a dramatic reduction in GCaMP3 fluorescence (*Figure 8G and H*). These data indicate that dFB sleep-promoting neurons inhibit OAA neurons.

## Discussion

A regular sleep pattern is achieved through coordinated action of neuronal signals reflecting internal sleep needs, the circadian clock and the level of arousal. In *Drosophila*, neurons projecting to the dorsal fan-shaped body of the central complex (dFB neurons) are proposed to be the effectors of the sleep homeostat (*Donlea et al., 2014*; *Donlea et al., 2018*; *Liu et al., 2016*; *Pimentel et al., 2016*). In our study, we identified multiple sleep and arousal-promoting neurons, which regulate the activity of dFB neurons through direct synaptic connections onto either their dendritic or axonal terminals. One of two groups of sleep-promoting neurons is a set of circadian pacemaker neurons, referred to as LPN[AstA] neurons. Thus, we propose that LPN[AstA] are key neurons that provide circadian input to the dFB to influence sleep. Moreover, we found that artificial co-activation of sleep and arousal-promoting neurons lead to frequent shifts between sleep and wake states, indicating that temporal segregation of the activities of these regulatory neurons is necessary to maintain consolidated sleep. An important issue for future investigation concerns the relationship between the normal activity patterns of LPN[AstA], dFB neurons and DAA neurons in live flies during sleep/wake conditions at different time during the circadian day/night cycles. The combination of highly sensitive, genetically encoded GCaMP sensors with an assay such as walking on an air supported ball should make this feasible (*Seelig et al., 2010*).

The NMDA receptor is proposed to promote sleep in *Drosophila* (*Robinson et al., 2016*; *Tomita et al., 2015*). Yet, the identity of the sleep-promoting glutaminergic synapse was unclear. Using the GRASP technique, we found that axons extending from both sets of sleep-promoting neurons (LPN[AstA] and SLP[AstA]) form close associations with dendrites of dFB neurons in the SMP region. Furthermore, we demonstrated that glutamate is required in the LPN[AstA] and SLP[AstA] neurons to promote sleep and activate dFB neurons. We used ATP to activate LPN[AstA] and SLP[AstA] neurons expressing the P2X2 channel and found that this manipulation lead to increased activity of dFB neurons. The combination of these GRASP and activity data suggest the existence of glutaminergic synapses between LPN[AstA]/SLP[AstA] and dFB neurons. However, our data do not distinguish which LPN[AstA]/SLP[AstA] neurons are glutamatergic.

We found that the LPN circadian pacemaker neurons express AstA—a neuropeptide, which has been linked to sleep promotion (*Chen et al., 2016*; *Donlea et al., 2018*). According to a previous model, AstA is released from dFB neurons since AstA is enriched at their axonal terminals, and an

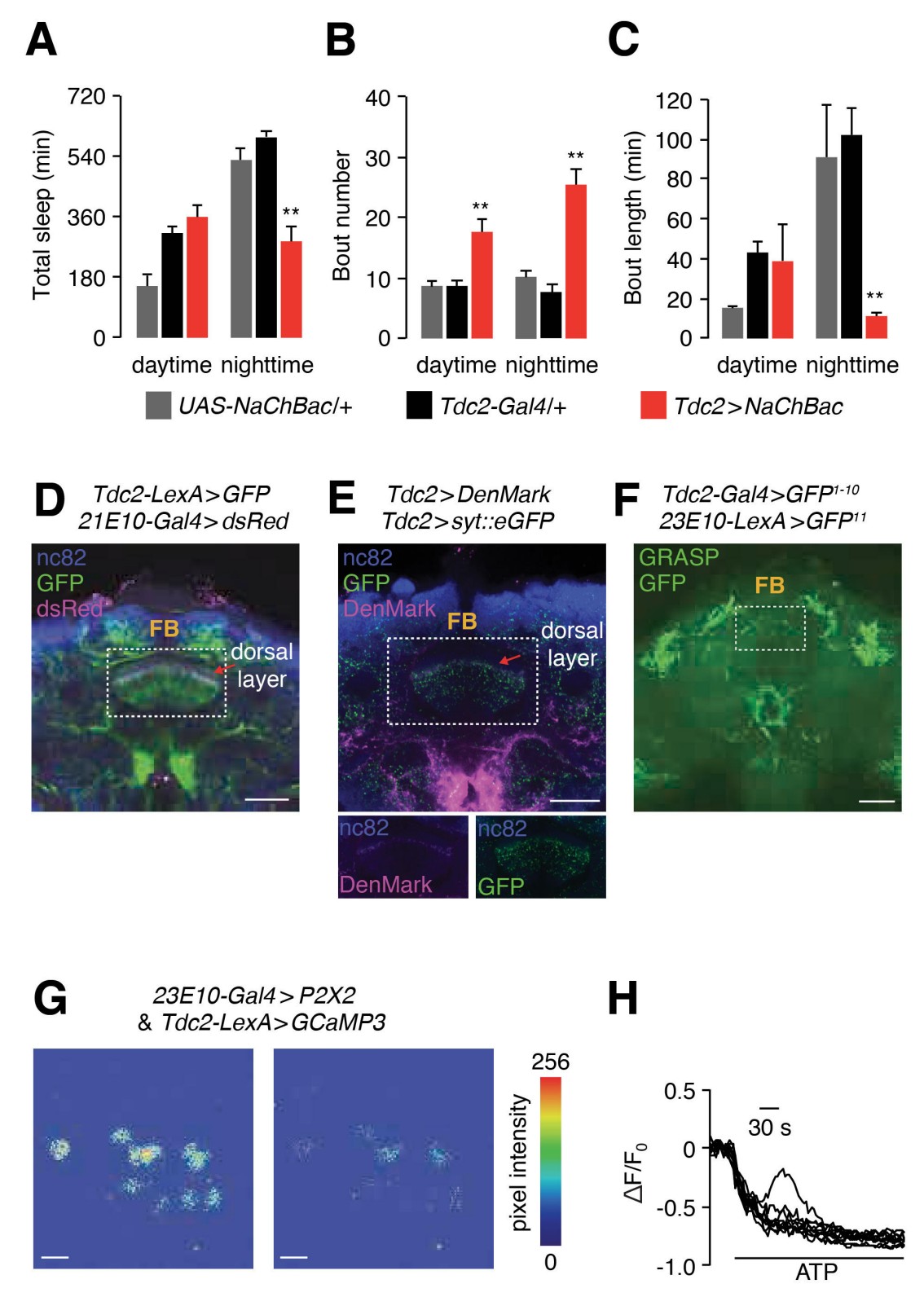

**Figure 8.** dFB neurons inhibit TDC2-positive arousal neurons. (**A—C**) Quantification of daytime and nighttime sleep parameters in flies expressing *UAS-NaChBac* under control of the *Tdc2-Gal4*. (**A**) Total sleep time. (**B**) Bout number. (**C**) Bout length. n = 24 for *UAS-NaChBac/+*, n = 16 for *Tdc2-Gal4/+*, n = 16 for *Tdc2 >NaChBac*. Error bars, SEMs. Unpaired Student's *t*-test. (**D**) A whole-mount brain co-expressing dFB and Tdc2 reporters (*23E10-Gal4 > dsRed* and *Tdc2-LexA > GFP*). The brain was co-stained with anti-dsRed and anti-GFP. The scale bar represents 40 μm. (**E**) Top, a whole-mount
*Figure 8 continued on next page*

*Figure 8 continued*

brain expressing *UAS-DenMark* and *UAS-syt-eGFP* under control of the *Tdc-Gal4*. The brain was co-stained with anti-dsRed (recognizes DenMark) and anti-GFP. The scale bar represents 40 μm. Bottom, images of the FB region showing the DenMark (anti-dsRed) and GFP signals separately. The boxed region contains the dFB. (**F**) Image of GRASP GFP fluorescence to test for close associations between Tdc2 and dFB neurons in the brain. The scale bar represents 60 μm. (**G**) Testing effect of GCaMP3 fluorescence in *Tdc2*-positive neurons after activating the P2X2 channel in dFB (*23E10-Gal4* positive) neurons with ATP. Representative images showing GCaMP3 fluorescence before and after bath application of 2.5 mM ATP. The scale bars represent 20 μm. (**H**) Representative traces showing the changes in GCaMP3 fluorescence (ΔF/F0) upon bath application of 2.5 mM ATP. The genotype is the same as in (**G**).

DOI: https://doi.org/10.7554/eLife.40487.016

RNAi transgene targeting *AstA* in dFB neurons was reported to recapitulate the sleep reduction exhibited by an *AstA* loss-of-function allele (*Donlea et al., 2018*). However, the anti-AstA signal appeared to juxtapose rather than overlap the axonal terminal of dFB neurons, and evidence that dFB neurons actually express the AstA peptide was not presented (*Donlea et al., 2018*). Thus, AstA may be released by other neurons near the terminals of dFB neurons. Consistent with this possibility, we found that both LPN[AstA] and SLP[AstA] neurons are labeled by AstA reporter lines, and LPN[AstA] neurons also express the AstA peptide. Our observations suggest that AstA released by LPN[AstA] and/or SLP[AstA] function as a sleep-promoting signal. Because dFB neurons are labeled by an *Allostatin-A Receptor 1* (*AstAR1*) reporter, the AstA released by LPN[AstA] and/or SLP[AstA] neurons might act directly on the AstAR1 in dFB neurons. This possibility is supported by our GRASP analyses indicating that LPN[AstA]/SLP[AstA] and dFB neurons are in close contact. AstAR1 has a role in promoting sleep (*Donlea et al., 2018*). However, it remains to be determined whether *AstAR1* is required in FB neurons for this sleep-promoting function.

Dopamine is an arousal neuromodulator in both fruit flies and mammals (*Andretic et al., 2008*; *Herrera-Solis et al., 2017*; *Liu et al., 2012*; *Ueno et al., 2012a*; *Ueno et al., 2012b*). In flies, dopamine can switch dFB neurons from an active to a quiescent state (*Pimentel et al., 2016*). In our study, we revealed a novel synaptic basis for dopaminergic regulation of dFB neurons. Our GRASP analyses reveal that dopaminergic arousal neurons (DAA) form close associations at various locations along the dendritic and axonal regions of dFB neurons. These include the SMP region where dFB neurons receive glutaminergic input. In addition, based on staining of dendritic and axonal makers, we propose that DAA neurons also form synapses with axons of dFB neurons in the dFB region. In support of this latter conclusion, we performed $Ca^{2+}$imaging and found that dopamine downregulates $Ca^{2+}$levels at the axonal terminals of dFB neurons. Thus, we propose that dopamine also contributes to arousal by suppressing the synaptic output from sleep promoting dFB neurons. dFB neurons receive both sleep-promoting glutaminergic and arousal-promoting dopaminergic inputs. Moreover, simultaneous activation of these opposing inputs by expressing *UAS-NaChBac* under control of the *TH-Gal4* and *65D05-Gal4* lines cause the flies to rapidly shift between sleep and wake states. Consequently, the animals are unable to experience long sleep bouts at night or stay awake for long periods during the day. These observations argue for a requirement for temporal segregation of input from sleep-promoting LPN[AstA] and SLP[AstA] neurons, and the arousal-promoting DAA neurons. While this approach enabled us to investigate the combined effects of sleep-promoting and arousal-promoting neurons on sleep it did not permit conclusions in terms of the precise dynamics of the shifts between sleep and wake states, as the *TH-Gal4* and *65D05-Gal4* lines presumably directs different levels of expression in the sleep- and arousal promoting neurons.

Our study also addressed the mechanism through which dFB neurons—the effectors of the sleep-homeostatic circuit, promote sleep. GABA is an inhibitory neurotransmitter and is known to promote sleep in flies and mammals (*Agosto et al., 2008*; *Chung et al., 2017*; *Jones, 2017*). We found that dFB neurons use GABA as their sleep-promoting neurotransmitter and that one of the downstream targets for dFB neurons are octopaminergic arousal-promoting (OAA) neurons. Based on GRASP analysis, we suggest that dFB neurons and OAA neurons form synapses. Because activating dFB neurons inhibits OAA neurons this inhibitory effect may serve to maintain long sleep bouts at night, since hyperactivation of OAA neurons causes fragmentation of nighttime sleep. Octopamine-producing neurons located in the medial protocerebrum are important for sleep (*Crocker et al., 2010*). However, it is unclear if these neurons are the OAA neurons downstream of dFB neurons.

Activation of dFB neurons inhibits the dopaminergic neurons that project to the mushroom bodies (*Berry et al., 2015*). This inhibition was proposed to impair forgetting. Arousal dopaminergic neurons project to the dFB, rather than the mushroom bodies. However, the soma of these dopaminergic arousal neurons are distributed in the same location as the dopaminergic neurons that mediate forgetting. Therefore, future experiments using a specific reporter that only mark the DAA neurons will be needed to address if activation of dFB neurons also inhibits DAA neurons. Finally, given that the dFB region is part of the central complex, which integrates multiple types of sensory information (*Wolff et al., 2015*), we propose that the GABA released by dFB neurons may provide inhibitory modulation that increases the threshold for sensory responses in sleeping animals.

# Materials and methods

## Key resources table

| Reagent type (species) or resource | Designation | Source or reference | Identifiers | Additional information |
|---|---|---|---|---|
| Genetic reagent (*D. melanogaster*) | UAS-NaChBac | Bloomington Drosophila Stock Center | stock # 9469; RRID:BDSC_9469 | |
| Genetic reagent (*D. melanogaster*) | UAS-TetxLC.tnt | Bloomington Drosophila Stock Center | stock# 28997; RRID:BDSC_28997 | |
| Genetic reagent (*D. melanogaster*) | UAS-TrpA1(B).K | Bloomington Drosophila Stock Center | stock# 26263; RRID:BDSC_26263 | |
| Genetic reagent (*D. melanogaster*) | UAS-GCaMP6f | Bloomington Drosophila Stock Center | stock# 42747; RRID:BDSC_42747 | |
| Genetic reagent (*D. melanogaster*) | VGlut-Gal80 | Bloomington Drosophila Stock Center | stock# 58448; RRID:BDSC_58448 | |
| Genetic reagent (*D. melanogaster*) | Tdc2-Gal4 | Bloomington Drosophila Stock Center | stock# 9313; RRID:BDSC_9313 | |
| Genetic reagent (*D. melanogaster*) | AstAR1-Gal4[23E10] | Bloomington Drosophila Stock Center | stock# 49032; RRID:BDSC_49032 | |
| Genetic reagent (*D. melanogaster*) | UAS-Shi[ts] | Bloomington Drosophila Stock Center | stock# 44222; RRID:BDSC_44222 | |
| Genetic reagent (*D. melanogaster*) | Gad1-RNAi | Bloomington Drosophila Stock Center | stock# 28079; RRID:BDSC_28079 | |
| Genetic reagent (*D. melanogaster*) | VMAT-RNAi | Bloomington Drosophila Stock Center | stock# 31257; RRID:BDSC_31257 | |
| Genetic reagent (*D. melanogaster*) | hs-FLP,UAS-mCD8::GFP | Bloomington Drosophila Stock Center | stock# 28832; RRID:BDSC_28832 | |
| Genetic reagent (*D. melanogaster*) | lexAOp2-CsChrimson.mVenus | Bloomington Drosophila Stock Center | stock# 55139; RRID:BDSC_55139 | |
| Genetic reagent (*D. melanogaster*) | UAS-CsChrimson.mVenus | Bloomington Drosophila Stock Center | stock# 55136; RRID:BDSC_55136 | |
| Genetic reagent (*D. melanogaster*) | UAS-DenMark, UAS-syt.eGFP | Bloomington Drosophila Stock Center | stock# 33064; RRID:BDSC_33064 | |
| Genetic reagent (*D. melanogaster*) | UAS-mCD8-GFP | Bloomington Drosophila Stock Center | stock# 5137; RRID:BDSC_5137 | |
| Genetic reagent (*D. melanogaster*) | lexAOp2-Gal80 | Bloomington Drosophila Stock Center | stock# 32217; RRID:BDSC_32217 | |
| Genetic reagent (*D. melanogaster*) | Tdc2-LexA | Bloomington Drosophila Stock Center | stock# 52242; RRID:BDSC_52242 | |
| Genetic reagent (*D. melanogaster*) | AstAR1-LexA[23E10] | Bloomington Drosophila Stock Center | stock# 53618; RRID:BDSC_53618 | |
| Genetic reagent (*D. melanogaster*) | AstA-LexA[65D05] | Bloomington Drosophila Stock Center | stock# 53625; RRID:BDSC_53625 | |
| Genetic reagent (*D. melanogaster*) | ChAT-RNAi | Bloomington Drosophila Stock Center | stock# 60028; RRID:BDSC_60028 | |

*Continued on next page*

*Continued*

| Reagent type (species) or resource | Designation | Source or reference | Identifiers | Additional information |
|---|---|---|---|---|
| Genetic reagent (*D. melanogaster*) | *VGlut-RNAi* | Bloomington Drosophila Stock Center | stock# 40927; RRID:BDSC_40927 | |
| Genetic reagent (*D. melanogaster*) | *UAS-Clk.Δ* | Bloomington Drosophila Stock Center | stock# 36318; RRID:BDSC_36318 | |
| Genetic reagent (*D. melanogaster*) | *AstA-Gal4$^{65D05}$* | Bloomington Drosophila Stock Center | stock# 39351; RRID:BDSC_39351 | |
| Genetic reagent (*D. melanogaster*) | *UAS-mCD8-RFP, LexAOp2-mCD8-GFP* | Bloomington Drosophila Stock Center | stock# 32229; RRID:BDSC_32229 | |
| Genetic reagent (*D. melanogaster*) | *TH-Gal4* | Bloomington Drosophila Stock Center | stock# 8848; RRID:BDSC_8848 | |
| Genetic reagent (*D. melanogaster*) | *UAS-FRT-CD2-stop-FRT-mCherry::trpA1* | PMID: 24860455 | | |
| Genetic reagent (*D. melanogaster*) | *UAS-P2X2, lexOp-GCaMP3* | PMID: 22539819 | | |
| Genetic reagent (*D. melanogaster*) | *UAS-cd4::spGFP1-10, lexAOp-cd4::spGFP11* | PMID: 19217375 | | |
| Genetic reagent (*D. melanogaster*) | *AstA$^{LexA}$* | This paper | | |
| Antibody | anti-dsRed (rabbit polyclonal) | Takara | Cat. 632496 | (1:1000) |
| Antibody | anti-GFP (rabbit polyclonal) | ThermoFisher Scientific | Cat. A11122 | (1:500) |
| Antibody | anti-GFP (chicken polyclonal) | ThermoFisher Scientific | Cat. A10262 | (1:1000) |
| Antibody | anti-GFP (mouse monoclonal) | Sigma-Aldrich | Cat. G6539 | (1:100) |
| Antibody | anti-Brp (termed as nc82, mouse monoclonal) | Developmental Studies Hybridoma Bank | DHSB: nc82 | (1:200) |
| Antibody | anti-Timeless (rat polyclonal) | other | | Obtained from Amita Sehgal's lab; (1:1000) |
| Antibody | anti-TH (rabbit polyclonal) | EMD Millipore | Cat. AB152 | (1:1000) |
| Antibody | anti-VGlut | other | | Obtained from Aaron Dianonio's lab; (1:1000) |
| Antibody | anti-AstA (mouse monoclonal) | Developmental Studies Hybridoma Bank | DHSB: 5F10 | (1: 50) |
| Antibody | Goat anti-mouse IgG1 | ThermoFisher Scientific | | Alexa Fluor 488, 568, 647; (1:1000) |
| Antibody | Goat anti-rat IgG | ThermoFisher Scientific | | Alexa Fluor 555; (1:1000) |
| Antibody | Goat anti-rabbit IgG | ThermoFisher Scientific | | Alexa Fluor 488, 568; (1:1000) |
| Antibody | Goat anti-chicken | ThermoFisher Scientific | | Alexa Fluor 488; (1:1000) |
| Recombinant DNA reagent | pBPLexA::p65Uw | Addgene | Cat. 26231 | pBPLexA::p65Uw was a gift from Gerald Rubin (Addgene plasmid # 26231; http://n2t.net/addgene:26231; RRID:Addgene_26231) |

*Continued on next page*

*Continued*

| Reagent type (species) or resource | Designation | Source or reference | Identifiers | Additional information |
|---|---|---|---|---|
| Commercial assay or kit | In-Fusion cloning kit | Takara | Cat. 121416 | |
| Chemical compound, drug | Mounting media | Vector Laboratories | Cat. H-1000 | |
| Chemical compound, drug | Triton X-100 | Sigma-Aldrich | Cat. T8787 | |
| Chemical compound, drug | L-glutamic Acid | Sigma-Aldrich | Cat. G1251 | |
| Chemical compound, drug | ATP | Sigma-Aldrich | Cat. 10519987001 | |
| Chemical compound, drug | Octopamine | Sigma-Aldrich | Cat. O0250 | |
| Chemical compound, drug | Tyramine | Sigma-Aldrich | Cat. T2879 | |
| Chemical compound, drug | Dopamine | Sigma-Aldrich | Cat. H8502 | |
| Software, algorithm | Sleeplab | other | | shared by Dr. William Joiner |
| Other | *Drosophila* Embryo Injection Services | BestGene Inc. | | |

## *Drosophila* culture conditions

Flies (*Drosophila melanogaster*) were cultured on cornmeal-agar-molasses medium under 12 hr light/12 hr dark cycles at room-temperature and ambient humidity. Detailed information regarding specific stains and genotypes is provided in the Key Resources Table section.

## Generation of *AstA$^{LexA}$* flies

To generate the *AstA$^{LexA}$* line with an insertion of the *LexA* reporter in place of endogenous AstA coding region (nucleotides 1–385 starting from endogenous start codon; *Figure 1G*), we used the CRISPR-HDR (clustered regularly interspaced short palindromic repeats – homology directed repair) method. We chose upstream (5') and a downstream (3') guide RNAs using the CRISPR Optimal Target Finder: http://tools.flycrispr.molbio.wisc.edu/targetFinder/. We annealed the following oligonucleotides to form two primer dimers. Each of the two primer dimers were cloned into the BbsI site of pU6-BbsI-ChiRNA (Addgene #45946) to generate two guide RNA expression plasmids: pU6-BbsI-ChiRNA-*AstA*_up, pU6-BbsI-ChiRNA-*AstA*_down

 *AstA*_up_ChiRNA_For: CTTCGCAGTAGGAGGTGGGCGTGA,
 *AstA*_up_ChiRNA_Rev: CGTCATCCTCCACCCGCACTCAAA
 *AstA*_down_ChiRNA_For: CTTCTACTTGGCAGCCGAGCGTGC
 *AstA*_down_ChiRNA_Rev: ATGAACCGTCGGCTCGCACGCAAA

We amplified the *AstA* upstream (1.1 kb) and downstream (1.1 kb) homology arms using the following primers:

 *AstA$^{LexA}$* upstream_For: CCGAAAAGTGCCACCTGACGTAACAAGTATCTGGAGGCA
 *AstA$^{LexA}$* upstream_Rev: CTTCATTTTGATTGCTAGCGAAAAAGTCAGCGAAGACA
 *AstA$^{LexA}$* downstream_For: ACATACTAGGCGCGCCCATATCTACTGTGTTCCTTCCTTT
 *AstA$^{LexA}$* downstream_Rev: ATGTCGACAAGCCGAACATACACATAAATTCTTAGACCATG

We used the In-Fusion cloning kit (Clontech) to insert the upstream and downstream homology arms into the KpnI and NdeI sites of pBPLexA::p65Uw (Addgene #26231), respectively (to create the pBPLexA::p65Uw-*AstA*_LA + RA plasmid). The pU6-BbsI-ChiRNA-*AstA*_up, pU6-BbsI-ChiRNA-*AstA*_down, and pBPLexA::p65Uw-*AstA*_LA + RA plasmids were injected into the BDSC #51323 strain, which provided the source of Cas9 (BestGene Inc.).

## Screening for RNAi lines that targets *VGlut, VMAT, ChAT and Gad1*

To screen for RNAi lines that lead to effective gene silencing upon expression in neurons, we screened a panel of available TRiP RNAi lines designed against *VGlut, VMAT, ChAT and Gad1*. We crossed these RNAi lines with *UAS-Dicer;;elav-Gal4* flies, which lead to pan-neuronal expression of these RNAi transgenes. We then screened for lines that either lead to lethality (line BL60028 for *ChAT* RNAi and line BL28079 for *Gad1* RNAi) or a large reduction of the corresponding gene products measured by quantitative RT-PCR or immunohistochemistry (*Figure 4—figure supplement 1A—F*)

## Sleep behavior

To measure sleep, we used 3—7 day-old female flies. Individual flies were loaded into glass tubes (provided with the Drosophila Activity Monitoring system (TriKinetics Inc.). The tubes each contained 5% sucrose and 1% agarose as the food source at one end and a small cotton plug at the other end. Flies were entrained for ≥2 days before activity data were collected for analysis. The activity data were collected in one-minute bins for further processing using MATLAB (MathWorks)-based software (*Koh et al., 2008*). If no activity was detected by the Drosophila Activity Monitoring system for more than 5 min, the fly was considered to be in a sleep state. Sleep assays were performed under white LED lights during the light period at 25°C unless otherwise indicated. For optogenetic stimulation, flies were cultured in regular food supplied with all-*trans* retinal in the dark for 24 hr before loading the animals into the glass tubes to perform activity measurements.

To test arousal, flies were maintained for 5 days under 12 hr light/12 hr dark cycles. On the 5th night, 5 min light pulses were delivered at ZT16, ZT18, and ZT20. To analyze the effects of the nighttime light stimuli on arousal, we quantified total sleep during the 30 min period beginning with the 5 min light stimuli and continuing an additional 25 min.

## Activation of neurons using optogenetics and thermogenetics

To measure sleep upon optogenetic stimulation, we used 3—7 day-old female flies. Flies were cultured in regular food supplied with all-*trans* retinal in the dark for 24 hr before loading individual flies into the same type of glass tubes mentioned above. The tubes contained 5% sucrose and 1% agarose as the food source at one end and a small cotton plug at the other end. Flies were entrained for 3 days under blue LED lights. On the 4th day, the LED lights were switched to red. Activity data were collected in one-minute bins for further processing using Sleep-Lab software. The effects on sleep as a result of optogenetic activation of neurons were quantified as the sleep change under red lights on day four verses blue lights on day 3.

We used the heat-activated channel, TRPA1, in combination with the *Gal4/UAS* system to activate neurons. To conduct these analyses, we introduced 3—7 day old female flies that were cultured on standard food into TriKinetics glass tubes with 5% sucrose +1% agarose as the food source at one end and a small cotton plug at the other end. Flies were entrained for 3 days under white LED lights at 22°C and the temperature was switched to 29°C on the day four to activate TRPA1. Activity data were collected in one-minute bins for further process using MATLAB-based software (*Joiner et al., 2013*). The effects on sleep as a result of thermogenetic manipulation of neuronal activity were quantified as the sleep change (in minutes) at 29°C (4th day) verses 22°C (3rd day).

## Whole-mount brain immunohistochemistry

For whole-mount brain immunohistochemistry, brains were dissected in phosphate buffered saline and 0.3% Triton X-100 (PBST). The brains were fixed with 4% paraformaldehyde (PFA) in PBST at room temperature for 20 min. Brains were briefly washed two times in PBST, blocked with PBST and 5% normal goat serum (NGS) at 4°C for 1 hr, and incubated overnight with primary antibodies diluted in PBST and 5% NGS at the following dilutions: 1) (1:1000) chicken anti-GFP, rabbit anti-dsRed, rabbit anti-TH antibody, and rat anti-Timeless, 2) (1:500) rabbit anti-GFP, 3) (1:200) nc82, 4) mouse anti-GFP antibody (1:100), and 5) (1:50) mouse anti-AstA. After three washes in PBST, we incubated the brains overnight in secondary antibodies corresponding to the primary antibodies used. Secondary antibodies were diluted 1:1000 in PBST and 5% NGS (refer to the Key Resource Table for the list of secondary antibodies). The brains were washed three times in PBST, and

mounted in VECTASHIELD (Vector Laboratories) on glass slides. Images were acquired with an upright Zeiss LSM 700 confocal microscope using 20X, 40X (oil) or 63X (oil) lenses.

## Mosaic expression of *mCherry::trpA1* using the *FlpOut* method

To use mCherry::TRPA1 to activate different subsets of neurons that express the *AstA-Gal4*, we conducted a mosaic analysis to express *mCherry::trpA1* in random subsets of *AstA-Gal4* neurons. To do so, we combined *hs-Flp* and *AstA-Gal4* transgenes with the *UAS-mCD8::GPF* transgene, and the *UAS-FRT-CD2-stop-FRT-mCherry::trpA1* transgene (*Vasmer et al., 2014*). The animals were maintained at room temperature (~22°C). To remove the *stop* cassette, we induced *hs-FLP* expression by heat shocking 3rd instar larval at 37°C for 1 hr. The animals were then returned to room temperature (~22°C) after the heat shock. We refer to these animals as *AstA >FlpOut-mCherry::trpA1* flies. We used 3—7 day-old female *AstA >FlpOut-mCherry::trpA1* flies to conduct the sleep measurements following the thermogenetic procedure described above. To tabulate the percentage of flies with a given amount of sleep change due to thermal activation of mCherry::TRPA1, we used the following calculation: sleep at 29°C minus sleep at 22°C.

To image the neurons, we dissected the brains of *AstA >FlpOut-mCherry::trpA1* flies that showed either sleep-promoting effects (sleep change >100 min) and that did not show sleep-promoting effects (sleep change <100 min). We then stained the brains with anti-GFP and anti-dsRed as described above (see Whole-mount Brain Immunohistochemistry).

## GRASP analyses

To conduct the GRASP analyses (*Feinberg et al., 2008*; *Gordon and Scott, 2009*), we expressed *lexAOp-CD4::spGFP11* and *UAS-CD4::spGFP1-10* under control of the *23E10 (AstAR1)-LexA* and either the *65D05 (AstA-Gal4)* or the *TH-Gal4*. To image GRASP signals, the brains were dissected in PBST, fixed with 4% PFA in PBST at room temperature for 20 min, and briefly washed with PBST before mounting in VECTASHIELD (Vector Laboratories) and viewing the fluorescence. To enhance GRASP signals, we performed immunostaining using mouse anti-GFP.

## Ca$^{2+}$ imaging in brains

To perform Ca$^{2+}$ imaging on whole-mount brains, expressing GCaMP3, GCaMP6f or syt::GCaMP6s, the flies were anaesthetized on ice and the brains were dissected into AHL buffer (108 mM NaCl, 5 mM KCl, 8.2 mM MgCl$_2$, 2 mM CaCl$_2$, 4 mM NaHCO$_3$, 1 mM NaH$_2$PO$_4$, 5 mM trehalose, 5 mM sucrose, and 5 mM HEPES pH 7.5), and transferred to an imaging chamber with a glass bottom, which was made from cover slides. The brains were immobilized with a metal harp and imaged using an inverted Zeiss LSM 800 confocal microscope. The regions of interest in the soma and neuronal processes were scanned in the time-lapse mode using 4—6 Z stacks. 30 or more cycles (60—90 s total) were imaged for checking the stability of the sample in the imaging chamber prior to adding chemicals to the bath. The average intensity during the last 10 of these 30 pre-cycles (before adding the chemicals) was used as the baseline (F) to calculate ΔF.

## Quantification and statistical analyses

No statistical method was applied to compute sample sizes when the study was designed. An appropriate sample size was adopted empirically according to published studies in the same field. No outlier data points were excluded in the statistical analyses. Animals were randomly selected into experimental groups. No masking was applied during group allocation, data collection and analysis.

To analyze sleep behavior, multiple animals were tested and the sleep index of individual animals were quantified independently. The numbers of animals tested were all biological replicates (each biological replicate was an independent individual animal) with numbers (n) indicated in all figure legends. For quantification of real-time PCR, each biological replicate was a sample of total RNA extracted from 10 animals of the indicated genotype. We used Student's *t*-tests to compare two group of samples. In most cases the tests were unpaired. We used paired Student's *t*-tests for comparing the effects of thermogenetic activation with TRPA1 in *Figure 7*. For comparing multiple groups, we used one-way ANOVA followed by the Dunnett test. Error bars indicate SEMs with indicated p values for statistical analysis. * Indicates p<0.05 and ** indicates p<0.01.

## Acknowledgments

The work was supported by grants to C M from the National Institute on Deafness and other Communication Disorders (DC007864) and the National Eye Institute (EY008117). We thank Drs. Orie Shafer and Julie Simpson for transgenic flies, Drs. Mark Wu and Sha Liu for guidance on performing sleep assays, Drs. Seghal and DiAntonio for antibodies, and Dr. William Joiner for sharing software. We also thank the Bloomington Drosophila Stock Center for providing fly stocks.

## Additional information

### Funding

| Funder | Grant reference number | Author |
| --- | --- | --- |
| National Institute on Deafness and Other Communication Disorders | DC007864 | Craig Montell |
| National Eye Institute | EY008117 | Craig Montell |

The funders had no role in study design, data collection and interpretation, or the decision to submit the work for publication.

### Author contributions

Jinfei D Ni, Conceptualization, Investigation, Data curation, Formal analysis, Visualization, Methodology, Writing—original draft, Writing—review and editing; Adishthi S Gurav, Weiwei Liu, Data curation, Investigation, Methodology; Tyler H Ogunmowo, Data curation, Investigation; Hannah Hackbart, Andrew A Verdegaal, Investigation; Ahmed Elsheikh, Data curation; Craig Montell, Conceptualization, Supervision, Funding acquisition, Investigation, Visualization, Project administration, Writing-original draft, Writing-review and editing

### Author ORCIDs

Jinfei D Ni  https://orcid.org/0000-0002-7004-1241
Weiwei Liu  http://orcid.org/0000-0001-5082-9114
Craig Montell  http://orcid.org/0000-0001-5637-1482

### Decision letter and Author response

Decision letter https://doi.org/10.7554/eLife.40487.019
Author response https://doi.org/10.7554/eLife.40487.020

## Additional files

### Supplementary files

• Transparent reporting form
DOI: https://doi.org/10.7554/eLife.40487.017

### Data availability

All data generated or analysed during this study are included in the manuscript and supporting files.

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
