## [Decision Letter]

Thank you for submitting your article "Differential regulation of the *Drosophila* sleep homeostat by circadian and arousal inputs" for consideration by *eLife*. Your article has been reviewed by three peer reviewers, including Mani Ramaswami as the Reviewing Editor and Reviewer #1, and the evaluation has been overseen by a Reviewing Editor and Eve Marder as the Senior Editor. The following individuals involved in review of your submission have agreed to reveal their identity: Alex C Keene (Reviewer #2); Paul Shaw (Reviewer #3).

The reviewers have discussed the reviews with one another and the Reviewing Editor has drafted this decision to help you prepare a revised submission.

Summary:

This is an exciting manuscript that further dissects the circuitry underlying sleep/wake regulation. Here, the authors map sleep and wake promoting neural circuitry that innervates the dorsal fan-shaped body, a critical regulator of sleep. In addition, the paper identifies neurotransmitters that function within this circuitry. The identification of LPN/SLP neurons as modulators of dFB function, and potential integrators of sleep and the circadian clock is a significant advance for the field, providing mechanistic insight into how sleep need is detected, and subsequently regulated. The authors also examine the role of dopamine neurons on sleep regulation, though this component of the manuscript is less novel. They further provide data indicating that dFB neurons release the neurotransmitter GABA to inhibit octopaminergic arousal neurons.

A strength of the manuscript are the complementary application of genetic manipulations and functional imaging to map sleep associated circuits. Overall, the manuscript is well written, with robust findings. The data are clearly presented, and their interpretation is supported by the data.

There are, however, a number of weaknesses most of which can be addressed without need for further experiments. The manuscript will be greatly improved by streamlining and moderating the presentation, with a greater effort to place the findings within the current literature and define sleep.

Essential revisions:

1) The current findings do not show the LPNs integrate sleep and the circadian clock. The fact that LPNs express clock genes does not necessarily indicate a role in circadian-regulation of sleep. How does perturbing a central clock component (Per/Tim or one another) in LPN^AstA^ neurons influence sleep? This is an obvious experiment to test and strengthen the interesting hypothesis that these neurons provide circadian input to sleep behavior.

2) The authors do not cite the original paper identifying the dFB as being sleep-promoting (Donlea et al., 2011). This oversight highlights a major concern that the authors do not show that the increased quiescence in *65D05>NaChBac* meets the criteria for sleep. In the original 2011 dFB paper, sleep was confirmed by examining arousal thresholds and sleep homeostasis as well as a host of other variables. But when new genetic manipulations are found to increase quiescence, one cannot simply assume that it is sleep, one must also determine whether the quiescence meets the other criteria. Arousal thresholds are relatively easy to quantify, and several labs have used a variety of techniques that do a nice job of measuring arousal thresholds. Sleep homeostasis is also a common tool used by a large number of *Drosophila* sleep labs and the authors should be able to quickly evaluate homeostasis. The DART system is a cheap and inexpensive way to monitor both. Quantifying arousal thresholds and homeostasis in the *65D05>NaChBac* flies and their parental controls. These analyses should be added to a revised manuscript.

3) The summary statement and description of the dFB in the text is slightly misleading. The literature suggests that it remains controversial whether dFB is “the” output arm of the homeostat (see Liu et al., 2016 and Seidner et al., 2015). It is likely one of many outputs, as evidence supports EB neurons, and non-central complex neurons are involved in homeostasis and sleep regulation. My understanding of the literature is there is not a singular sleep center, and it is likely regulated (in flies and mammals) by dispersed circuit connectivity.

4) The data presentation should be improved and better aligned with the existing literature. Traditionally, sleep traces over the 24 hour day for both parental controls (*UAS/+* and *Gal4/+* are plotted on the same graph as the experimental line (*Gal4>UAS*) so that they can be directly compared. However, the authors have decided to only show one genotype on each graph. This approach makes it very difficult to determine exactly what happened. Moreover, the authors don't show all of the parental lines in the main text. It turns out that the only way for a reader to fully understand how the experimental line behaves is to compare it directly to both parental lines. Are the sleep traces depicted for the experimental lines and parental lines from the same experiment? Do the experimental lines show an odd pattern of sleep? The data would be much easier to understand and interpret if the authors compared the experimental line directly to both parental lines in the same graph.

5) In the same vein, it is worth noting, the authors do plot multiple genotypes on the same graph when they want to emphasize a difference (e.g. Figure 6A). However, the authors are selective in which parental line to us. For example, the authors show the *UAS-NaChBac*/+ controls in Figure 6 without showing the *Gal4*/+ for comparison but show the *Tdc2-Gal4*/+ in Figure 8 without showing the *UAS-NaChBac*/+. Was sleep only evaluated once for the *UAS-NaChBac*/+ group? Plotting the full experiment together eliminates these questions. Indeed, in Figure 6, *65D05*/+ and *TH-Gal4*/+ seem to be missing from the analysis. It is inappropriate and misleading to make a comparison between an experimental line and just one of the two parental lines. Both comparisons are required.

6) Regarding the co-activation experiments, there are a number of implicit assumptions in how this approach may or may not work. The interpretation of the dataset may not be as straightforward as presented. The manuscript would be greatly improved if the authors devote space in the Discussion section on the strengths and pitfalls of this approach. For example, is it obvious that *TH-Gal4* results in the same level of Gal4 expression as *65D05-Gal4*? If the expression levels are different, how would this alter the interpretation of the results?

7) The images shown throughout the manuscript are very nice. Is there a reason that we don't see the merge for Figure 2D-I?

8) How closely does the expression patter of the *65D05-LexA* driver match the expression pattern of *65D05-Gal4*? This should be shown and described.

9) Overall, the imaging experiments and circuit experiments rely on ectopic manipulation of neural activity. This means limited inferences can be drawn about the activity of these circuits under endogenous sleep-wake conditions. With current behavioral technology (for example, fly-ball trackers) this can be addressed. It would be useful to comment on this in the Discussion section.

10) As an extension of the previous point, changes in dFB when modulated putatively upstream neurons to not mean these neurons are directly acting on the dFB, but simply are part of the circuit.

11) The presentation of Figure 1J is confusing. It may be improved by adding genotypes above the graphs (as in Figure 1A-C) and using different colors to depict light protocol from those used to depict genotype.

12) Are LPN^AstA^ and SLP^AstA^ neurons glutamatergic and is *VGlut* required in LPN^AstA^ cells? The limitations of the existing data should be acknowledged or extended.

13) Is AstA required in these AstA neuronal populations? If this is not known, then the issue should be addressed or formally acknowledged.

14) Similarly, is AstAR1 required in FB neurons?

15) Is there a really a population of AstA expressing neurons adjacent to sleep promoting, AstA-receptor expressing FB neurons in the fan-shaped body as suggested? Instead of using Gal4 or reporter lines, can these issues be addressed using antibody staining? Do these neurons play any role in sleep control? This should be discussed.

16) Has previous work shown that inhibition of OAA neurons increases arousal threshold? This is implied as the likely reason for FB>OAA connectivity. Also, is it known how OAA neurons connect to the FB? Please clarify.

17) Does FB neuron activation also inhibit DAA neurons? Is this known and/or can it be determined or discussed?

18) The authors cite Berry et al. (2015) in relation to the role of the dFB and sleep homeostasis but the focus of the paper was on forgetting.

19) The authors cite Hendricks et al. (2000) for determining that a 5 minute bin was used to define sleep even though the group used a 30 minute bin until 2003; the first mention of a 5 minute bin was in 2000.

---

## [Author Response]

Essential revisions:1) The current findings do not show the LPNs integrate sleep and the circadian clock. The fact that LPNs express clock genes does not necessarily indicate a role in circadian-regulation of sleep. How does perturbing a central clock component (Per/Tim or one another) in LPN^AstA^ neurons influence sleep? This is an obvious experiment to test and strengthen the interesting hypothesis that these neurons provide circadian input to sleep behavior.

To test for a potential contribution of the central clock for the sleep-promoting function of LPN^AstA^ neurons we expressed a dominate-negative isoform of the central clock component, Clock (*UAS-Clk∆*) in LPN^AstA^ neurons under control of the *65D05-Gal4*. We found that the sleep-promoting effect due to neuronal hyperactivation of LPN^AstA^ neurons (*65D05-Gal4>NaChBac*) during the daytime was reduced by expression of *Clk∆* (Figure 4—figure supplement 2A and B). This indicates that the central clock in LPN^AstA^ neurons contributes to the sleep-promoting function of these neurons. We describe these data in subsection “Sleep-promoting role of AstA and a central clock component in LPN^AstA^ neurons”.

2) The authors do not cite the original paper identifying the dFB as being sleep-promoting (Donlea et al., 2011). This oversight highlights a major concern that the authors do not show that the increased quiescence in 65D05>NaChBac meets the criteria for sleep. In the original 2011 dFB paper, sleep was confirmed by examining arousal thresholds and sleep homeostasis as well as a host of other variables. But when new genetic manipulations are found to increase quiescence, one cannot simply assume that it is sleep, one must also determine whether the quiescence meets the other criteria. Arousal thresholds are relatively easy to quantify, and several labs have used a variety of techniques that do a nice job of measuring arousal thresholds. Sleep homeostasis is also a common tool used by a large number of Drosophila sleep labs and the authors should be able to quickly evaluate homeostasis. The DART system is a cheap and inexpensive way to monitor both. Quantifying arousal thresholds and homeostasis in the 65D05>NaChBac flies and their parental controls. These analyses should be added to a revised manuscript.

We added the Donlea et al. (2011) reference in the first paragraph of the Introduction. To test the effects of activating dFB neurons on arousal threshold we maintained the flies for five days under 12 hour light/12 hour dark cycles. On the fifth night, we exposed the flies to three light-pulses (5-min each delivered at ZT16, ZT18, and ZT20; Figure 1—figure supplement 1A). In control flies (*65D05-Gal4/+* and *UAS-NaChBac/+),* the light stimuli caused large reductions in sleep during the 30-minute period following the onset of the light stimuli (Figure 1—figure supplement 1B-D). However, the *65D05>NaChBac* flies, did not exhibit decreased sleep (arousal) in response to the 5 minute light stimulations (Figure 1—figure supplement 1B and E). These data support the proposal thathyperactivationof dFB neurons increases sleep. We described these new results in the Results section.

3) The summary statement and description of the dFB in the text is slightly misleading. The literature suggests that it remains controversial whether dFB is “the” output arm of the homeostat (see Liu et al., 2016 and Seidner et al., 2015). It is likely one of many outputs, as evidence supports EB neurons, and non-central complex neurons are involved in homeostasis and sleep regulation. My understanding of the literature is there is not a singular sleep center, and it is likely regulated (in flies and mammals) by dispersed circuit connectivity.

In response to this comment, we slightly modified the first sentence of the abstract from “*The* output arm of the sleep homeostat in *Drosophila is* a group of neurons with projections to the dorsal fan-shaped body (dFB neurons) of the central complex in the brain” to “*One* output arm of the sleep homeostat in *Drosophila appears to be* a group of neurons with projections to the dorsal fan-shaped body (dFB neurons) of the central complex in the brain.”

In addition, we added the following sentence to the Introduction:

“Other non-central complex neurons are also involved in sleep regulation and different neuronal pathways elicit distinct effects on the sleep homeostat (Seidner et al., 2015).”

4) The data presentation should be improved and better aligned with the existing literature. Traditionally, sleep traces over the 24 hour day for both parental controls (UAS/+ and Gal4/+ are plotted on the same graph as the experimental line (Gal4>UAS) so that they can be directly compared. However, the authors have decided to only show one genotype on each graph. This approach makes it very difficult to determine exactly what happened. Moreover, the authors don't show all of the parental lines in the main text. It turns out that the only way for a reader to fully understand how the experimental line behaves is to compare it directly to both parental lines. Are the sleep traces depicted for the experimental lines and parental lines from the same experiment? Do the experimental lines show an odd pattern of sleep? The data would be much easier to understand and interpret if the authors compared the experimental line directly to both parental lines in the same graph.

The experimental and parental lines were conducted in the same experiments. We have now added combined graphs to simplify comparisons, although in most cases we also include the separate graphs so that the individual graphs can be viewed with clarity.

5) In the same vein, it is worth noting, the authors do plot multiple genotypes on the same graph when they want to emphasize a difference (e.g. Figure 6A). However, the authors are selective in which parental line to us. For example, the authors show the UAS-NaChBac/+ controls in Figure 6 without showing the Gal4/+ for comparison but show the Tdc2-Gal4/+ in Figure 8 without showing the UAS-NaChBac/+. Was sleep only evaluated once for the UAS-NaChBac/+ group? Plotting the full experiment together eliminates these questions. Indeed, in Figure 6, 65D05/+ and TH-Gal4/+ seem to be missing from the analysis. It is inappropriate and misleading to make a comparison between an experimental line and just one of the two parental lines. Both comparisons are required.

We added analyses of the *65D05-Gal4*/+ and the *TH-Gal4*/+ to Figure 6. We added the *UAS-NaChBac*/+ control to Figure 8.

6) Regarding the co-activation experiments, there are a number of implicit assumptions in how this approach may or may not work. The interpretation of the dataset may not be as straightforward as presented. The manuscript would be greatly improved if the authors devote space in the Discussion section on the strengths and pitfalls of this approach. For example, is it obvious that TH-Gal4 results in the same level of Gal4 expression as 65D05-Gal4? If the expression levels are different, how would this alter the interpretation of the results?

This is a valid point. To alert the reader to the limitation of this approach, we added the following point to the Discussion section:

“While this approach enabled us to investigate the combined effects of sleep-promoting and arousal-promoting neurons on sleep it did not permit conclusions in terms of the precise dynamics of the shifts between sleep and wake states, as the *TH-Gal4* and *65D05-Gal4* lines presumably directs different levels of expression in the sleep- and arousal promoting neurons.”

7) The images shown throughout the manuscript are very nice. Is there a reason that we don't see the merge for Figure 2D-I?

We added the merge images to Figure 2.

8) How closely does the expression patter of the 65D05-LexA driver match the expression pattern of 65D05-Gal4? This should be shown and described.

To address this question, we performed double-labeling experiments using *lexAop-mCD8::GFP* and *UAS-mCD8::RFP* expressed under control of the *65D05-Gal4* and *65D05-LexA*, respectively. We found that the two reporters largely labeled the same sets of neurons in the fly brain (Figure 3—figure supplement 2). Of particular importance to our work, both the *LexA* and *Gal4* reporters labeled the SLP^AstA^ (Figure 3—figure supplement 2A-C) and LPN^AstA^ neurons (Figure 3—figure supplement 2D-F and the Results section).

9) Overall, the imaging experiments and circuit experiments rely on ectopic manipulation of neural activity. This means limited inferences can be drawn about the activity of these circuits under endogenous sleep-wake conditions. With current behavioral technology (for example, fly-ball trackers) this can be addressed. It would be useful to comment on this in the Discussion section.

We added this point to the Discussion section.

10) As an extension of the previous point, changes in dFB when modulated putatively upstream neurons to not mean these neurons are directly acting on the dFB, but simply are part of the circuit.

We suggest the combination of activating upstream neurons combined with GRASP experiments allow us to “suggest” direct interactions. We expanded the second paragraph of the Discussion section to make this point.

“The NMDA receptor is proposed to promote sleep in *Drosophila* (Robinson et al., 2016; Tomita et al., 2015). […] The combination of these GRASP and activity data suggest the existence of glutaminergic synapses between LPN^AstA^/SLP^AstA^ and dFB neurons.”

11) The presentation of Figure 1J is confusing. It may be improved by adding genotypes above the graphs (as in Figure 1A-C) and using different colors to depict light protocol from those used to depict genotype.

We modified Figure 1 as suggested.

12) Are LPN^AstA^ and SLP^AstA^ neurons glutamatergic and is VGlut required in LPN^AstA^ cells? The limitations of the existing data should be acknowledged or extended.

Our data support the conclusion that at least some LPN^AstA^ and SLP^AstA^ neurons are glutamatergic. Specifically, we found that the sleep-promoting effect due to expression of NaChBac in LPN^AstA^ and SLP^AstA^ neurons (*65D05-Gal4>NaChBac*) is suppressed by RNAi knock down of the vesicular glutamate transporter (*VGlut;* Figure 4A, E and F). However, our data do not distinguish which LPN^AstA^/SLP^AstA^ neurons are glutamatergic. We now acknowledge this limitation in the Discussion section.

13) Is AstA required in these AstA neuronal populations? If this is not known, then the issue should be addressed or formally acknowledged.

To address whether AstA functions in LPN^AstA^ neurons to promote sleep, we conducted additional experiments. We examined whether removal of AstA impacted on the sleep-promoting effect induced by hyperactivation of these neurons (*65D05-Gal4>NaChBac*). We found that the *AstA^LexA^* mutation significantly decreased the sleep-promoting effect caused by expression of NaChBac (Figure 4—figure supplement 2C and D). The reduction in sleep due to the *AstA^LexA^* mutation was most pronounced during the second half of the daytime period. These data are now described in subsection” Sleep-promoting role of AstA and a central clock component in LPN^AstA^ neurons”.

14) Similarly, is AstAR1 required in FB neurons?

Unfortunately, the reagents necessary to address this question are not available to us and we could not obtain them in a timely fashion. We added the following to the Discussion section to clarify this point:

“Because dFB neurons are labeled by an *Allatostatin-A Receptor 1 (AstAR1*) reporter, the AstA released by LPN^AstA^ and/or SLP^AstA^ neurons might act directly on the AstAR1 in FB neurons. Moreover, AstAR1 has a role in promoting sleep (Donlea et al., 2018). However, it remains to be determined whether *AstAR1* is required in FB neurons for this sleep-promoting function.”

15) Is there a really a population of AstA expressing neurons adjacent to sleep promoting, AstA-receptor expressing FB neurons in the fan-shaped body as suggested? Instead of using Gal4 or reporter lines, can these issues be addressed using antibody staining? Do these neurons play any role in sleep control? This should be discussed.

AstA-receptor antibodies are not available. AstA antibodies are available and stain LPN^AstA^ neurons (Figure 2J). However, the projections of LPN^AstA^ neurons are visualized more clearly with the *AstA^LexA^* reporter (*AstA^LexA^>GFP*). These projections extend to the SMP region (Figure 3G) and form potential synapses with dFB neurons, which are labeled by the AstAR1 reporter, *23E10-Gal4*. Thus, these AstA and AstAR1 neurons are potentially adjacent. This proposal is further supported by the results of our GRASP analysis (Figure 3J). This is described in the Discussion section:

“…we found that both LPN^AstA^ and SLP^AstA^ neurons are labeled by AstA reporter lines, and LPN^AstA^ neurons also express the AstA peptide. Our observations suggest that AstA released by LPN^AstA^ and/or SLP^AstA^ function as a sleep-promoting signal. Because dFB neurons are labeled by an *Allatostatin-A Receptor 1 (AstAR1*) reporter, the AstA released by LPN^AstA^ and/or SLP^AstA^ neurons might act directly on the AstAR1 in FB neurons. This possibility is supported by our GRASP analyses indicating that LPN^AstA^/SLP^AstA^ and dFB neurons are in close contact.”

16) Has previous work shown that inhibition of OAA neurons increases arousal threshold? This is implied as the likely reason for FB>OAA connectivity. Also, is it known how OAA neurons connect to the FB? Please clarify.

Previous experiments on OAA neurons focused on activation rather than inhibition of OAA neurons. This manipulation reduced sleep and caused a consequent reduction in arousal threshold (Crocker and Sehgal, 2010). Presumably, inhibition of OAA neurons would increase arousal threshold. However, this has not been demonstrated experimentally. In the Discussion section, we comment on the implication of the FB to OAA connectivity:

“Because activating dFB neurons inhibits OAA neurons, this inhibitory effect may serve to maintain long sleep bouts at night since hyperactivation of OAA neurons causes fragmentation of nighttime sleep. Octopamine-producing neurons located in the medial protocerebrum are important for sleep (Crocker et al., 2010). However, it is unclear if these neurons are the OAA neurons downstream of dFB neurons.”

17) Does FB neuron activation also inhibit DAA neurons? Is this known and/or can it be determined or discussed?

Berry et al. (2015) found that FB neurons inhibit DAA neurons that project to the mushroom bodies, and they propose that this inhibition impairs dopamine-mediated forgetting. Arousal DAA neurons project to the FB, rather than the mushroom bodies. However, the soma of these dopaminergic arousal neurons are distributed in the same location as the dopaminergic neurons that mediate forgetting. Therefore, future experiments using a specific reporter that only mark the DAA neurons will be needed to address if FB neuron activation also inhibits DAA neurons. We modified our Discussion section.

18) The authors cite Berry et al. (2015) in relation to the role of the dFB and sleep homeostasis but the focus of the paper was on forgetting.

We changed the citation of Berry et al. (2015) to Donlea et al. (2011).

19) The authors cite Hendricks et al. (2000) for determining that a 5 minute bin was used to define sleep even though the group used a 30 minute bin until 2003; the first mention of a 5 minute bin was in 2000.

We made the correction.